# Northwestward Cropland Expansion and Growing Urea-Based Fertilizer Use Enhanced NH₃ Emission Loss in the Contiguous United States

Peiyu Cao, Chaoqun Lu, Jien Zhang, Avani Khadilkar

Department of Ecology, Evolution, and Organismal Biology, Iowa State University, Ames, Iowa, USA

*Correspondence to*: Chaoqun Lu (clu@iastate.edu)

**Abstract.** The increasing demands of food and biofuel have promoted cropland expansion and nitrogen (N) fertilizer enrichment in the United States over the past century. However, the role of such long-term human activities in influencing the spatiotemporal patterns of Ammonia (NH₃) emission remains poorly understood. Based on an empirical model and time-series gridded data sets including temperature, soil properties, N fertilizer management, and cropland distribution history, we have quantified monthly fertilizer-induced NH₃ emission across the contiguous U.S from 1900 to 2015. Our results show that N fertilizer-induced NH₃ emission in the U.S. has increased from $< 50$ Gg N yr$^{-1}$ before the 1960s to 641 Gg N yr$^{-1}$ in 2015, for which corn and spring wheat are the dominant contributors. Meanwhile, urea-based fertilizers gradually grew to the largest NH₃ emitter and accounted for 78% of the total increase during 1960-2015. The factorial contribution analysis indicates that the rising N fertilizer use rate dominated the NH₃ emission increase since 1960, whereas the impacts of temperature, cropland distribution and rotation, and N fertilizer type varied among regions and over periods. Geospatial analysis reveals that the hotspots of NH₃ emissions have shifted from the central U.S. to the northwestern U.S. from 1960 to 2015. The increasing NH₃ emissions in the northwestern U.S has been found to closely correlate to the elevated NH₄⁺ deposition in this region over the last three decades. This study shows that April, May, and June account for the majority of NH₃ emission in a year. Interestingly, the peak emission month has shifted from May to April since the 1930s. Our results imply that the northwestward corn and spring wheat expansion and growing urea-based fertilizer uses have dramatically altered the spatial pattern and temporal dynamics of NH₃ emission, impacting air pollution and public health in the U.S.

## 1 Introduction

The tremendous increase in synthetic nitrogen (N) fertilizer uses has greatly promoted crop yields in the U.S. since the early 20[th] century (Cao et al., 2018; Erisman et al., 2008). The predictable rise in food demand may lead to more N fertilizer consumption in the coming decades (Alexandratos and Bruinsma, 2012; David et al., 1997). However, 5-9% of the N applied was lost to the atmosphere through ammonia (NH₃) volatilization (0.5-1 Tg N annually) across the U.S. at the beginning of this century, which lowered the N use efficiency (NUE) of crops and caused numerous environmental issues (Bouwman et al., 2002; Cassman and Walters, 2002; Lu et al., 2019; Tilman et al., 2002). Nationwide, synthetic N fertilizer-induced NH₃

volatilization, contributing to 15-30% of annual total $NH_3$ emission, has been identified as the second contributor to atmospheric $NH_3$ only next to livestock production (Park et al., 2004; Paulot et al., 2014; Reis et al., 2009; U.S. EPA, 2019). Atmospheric $NH_3$ plays a significant role in the formation of atmospheric particulate matters (PM) and is an important component of N deposition (Behera et al., 2013), which can degrade visibility, induce respiratory and cardiovascular disease, cause eutrophication of aquatic ecosystems, soil acidification, and reduce biodiversity (Bowman et al., 2008; Galloway et al., 2003). Thus, to quantify fertilizer-derived $NH_3$ emission over space and time is essential in assessing agricultural N budget and improving the accuracy of air quality modeling (Eickhout et al., 2006; Gilliland et al., 2006; Van Grinsven et al., 2015).

However, it is challenging to quantify fertilizer-induced $NH_3$ emissions due to the paucity of information on spatially and temporally varied environmental conditions and various agricultural practices (Behera et al., 2013; Bouwman et al., 2002; Pinder et al., 2006; Sommer et al., 2004). Inverse modeling of atmospheric observations such as N deposition and satellite images has been developed as an indirect approach to estimate seasonal $NH_3$ emissions at the regional and national scale (Gilliland et al., 2006; Liu et al., 2019)). However, this "top-down" approach has difficulty in separating the contribution of each source of $NH_3$ emission. Process-based modeling is a popular "bottom-up" approach for quantifying spatially explicit $NH_3$ emissions over a long period (Cooter et al., 2012; Riddick et al., 2016; Xu et al., 2018). These models require detailed information on local environmental conditions and farming practices that is generally not available. Besides, background emissions (i.e., prior to human disturbances) are always included in such modeling estimations. Another widely-used "bottom-up" approach to estimate the single source of $NH_3$ emission is by emission factor (EF), which represents the proportion of $NH_3$ volatilization from N input. Compared to the constant EFs used to estimate annual $NH_3$ emissions in early studies, more recent empirical estimations have been improved to provide seasonal estimations based on environmental conditions and agricultural management practices (Bouwman et al., 2002; Goebes et al., 2003; Huang et al., 2012; Jiang et al., 2017; Kang et al., 2016). For example, by considering three fertilizer application timings and adopting EFs that consider differences among crop types, environmental factors, and fertilizer types, Paulot et al. (2014) estimated monthly $NH_3$ emission from N fertilizer uses in the U.S. during 2005-2008.

While more recent "top-down" and "bottom-up" estimations have quantified the seasonality and spatial heterogeneity of $NH_3$ emissions across the country during a short period, few studies have assessed the spatiotemporal patterns and the factorial contributions of $NH_3$ emissions on a century scale. The lack of long-term assessment and understanding of contributing factors may limit our capability in predicting the dynamics of $NH_3$ emission under future changes in climate, land use, and agricultural management practices (Zhu et al., 2015). The hotspots of intensive agricultural cultivation and N fertilizer uses have shifted from the southeast U.S. to the Midwest and Northern Great Plains during the 20th Century (Cao et al., 2018; Johnston, 2014; Nickerson et al., 2011; Yu et al., 2018; Yu and Lu, 2018). It is reported that land sources of $NH_3$ play an important role in affecting the atmospheric N deposition and $PM_{2.5}$ (Du et al., 2014; Li et al., 2016; U.S. EPA, 2019), but it remains less known how land use change and N fertilizer management history have altered $NH_3$ emissions since 1900.

Based on spatially explicit time-series of cropland distribution maps and N fertilizer management database, we adopted empirical modeling of EF to calculate monthly $NH_3$ emissions from synthetic N fertilizer uses (Hereafter, $NH_3$ emission refers

to the synthetic N fertilizer-induced NH$_3$ emission unless specified otherwise) in the contiguous U.S. at a resolution of 1 km $\times$ 1 km from 1900 to 2015. We examined the magnitude, spatiotemporal pattern, and seasonality of NH$_3$ emissions at national and regional scales driven by changes in historical temperature, land use, and N fertilizer management practices. Our goals are to answer the following questions: (1) how did the NH$_3$ emission change over space and time? (2) what roles did temperature, land use, crop rotation, and N fertilizer use play in determining the changes in NH$_3$ emission? (3) what is the relationship between atmospheric NH$_4^+$ deposition dynamics and NH$_3$ emission at seasonal and annual bases?

## 2 Materials and Methods

In this study, we used a widely-used residual maximum likelihood model (REML, Bouwman et al., 2002) derived emission factor (EF) to assess synthetic N fertilizer-induced NH$_3$ emissions. We calculated the REML-emission factors based on spatial datasets of air temperature, soil properties, crop type, N fertilizer type and application method at a resolution of 1 km $\times$1 km. Our recent work has reconstructed the U.S. state-level crop-specific N fertilizer management history with information of application timing, application method, and fertilizer types from 1900 to 2015 (Cao et al., 2018). In this study, we assigned N fertilizer use rates into exact days each year by linking fertilizer application timings with a harmonized database of crop phenology dates. The daily fertilizer input rate was then aggregated to monthly time step. We spatialized the monthly N fertilizer use data generated above to the U.S. 1-km gridded cropland distribution maps developed by Yu and Lu (2018). By multiplying N fertilizer use rates with EF, we obtained spatially explicit estimates of NH$_3$ emission at a monthly time step from 1900 to 2015. For display purposes, we resampled the spatial time-series of NH$_3$ emissions to 5 km $\times$ 5 km resolution with the average NH$_3$ emission depicted in each pixel. To represent the regional difference of NH$_3$ emission and its impact on N deposition, we partitioned the entire contiguous U.S. into seven regions: the Northwest (NW), the Southwest (SW), the Northern Great Plains (NGP), the Southern Great Plains (SGP), the Midwest (MD), the Southeast (SE), and the Northeast (NE) according to the U.S. Fourth National Climate Assessment (2019) (Fig. 2).

### 2.1 REML Model

Bouwman et al. (2002) summarized 1667 NH$_3$ volatilization measurements in 148 research papers to assess the effects of a variety of human management practices and environmental factors on NH$_3$ emission at a global scale. Finally, six factors including air temperature, soil pH, soil Cation Exchange Capacity (CEC), crop type, fertilizer type, and application method, are considered in the REML model to determine the EF and then calculate the NH$_3$ emission (Eq. 1, Eq. 2).

$$EF = exp(FV_{Tem}+FV_{pH} + FV_{CEC} + FV_{FT} + FV_{AM} + FV_{CT}) \tag{1}$$

$$NH_3 = N\,Fer * EF \tag{2}$$

Where in Eq. 1, *EF* refers to emission factor. *FV* refers to Factor Value of each key driver. The values of input data are grouped into broad classes. *Tem* refers to air Temperature and are grouped into two classes above and below 20 ℃. *pH* refers to soil pH and has four classes. *CEC* refers to soil CEC and has four classes. *FT* refers to fertilizer type, including 12 types. *AM* refers to N fertilizer application method, including broadcast, incorporate, solution, broadcast and then flood, and incorporate and then flood. *CT* refers to Crop Type and is classified as Upland crops, Grass, and Flooded crops. Where in Eq. 2, $NH_3$ refers to $NH_3$ emission and *N fer* refers to N fertilizer use rate. More detailed grouping information and the corresponding factor value can be found in the supplementary Table S1.

## 2.2 Input data preparation

### 2.2.1 Temperature

We downloaded the daily temperature data from high-resolution gridded data products TS 2.1 from station observations by the Climatic Research Unit (CRU) of the University of East Anglia TS 2.1 and North America Regional Reanalysis (NARR) dataset from a combination of modeled and observed data (Mesinger et al., 2006; Mitchell and Jones, 2005). The daily temperature data were further resampled to 1km × 1km and aggregated to monthly average temperature.

### 2.2.2 Soil pH and CEC

We resampled the soil properties data (pH and CEC) obtained from Geospatial Data Gateway (gSSURGO, 2018) to 1 km ×1 km resolution. Among a variety of measurements, we adopted the soil pH from *ph1to1h2o_r* in the attribute table, which uses the negative logarithm at base 10 of the hydrogen ion activity in the soil using the 1:1 soil-water ratio method. Meanwhile, we chose *cec7_r* in the attribute table as our soil CEC indicator, which represents the amount of readily exchangeable cations that can be electrically absorbed to negative charges in the soil, soil constituent, or other material at pH 7.0, as estimated by the ammonium acetate method.

### 2.2.3 Cropland distribution maps

We adopted a newly developed cropland distribution and type maps of the contiguous U.S. at a resolution of 1 km × 1 km from 1900 to 2015 to drive the REML model (Yu et al., 2018; Yu and Lu, 2018). The cropland maps were reconstructed to characterize the area, type, and distribution of cultivated land annually by harmonizing various sources of inventory data and remote sensing images. By using this database, we identified and tracked the percent of cropped land area, and what crop was planted in each grid cell each year while excluding summer idle/fallow areas. Ten major crop types identified in the cropland maps and used in this study include corn, soybean, winter wheat, spring wheat, cotton, sorghum, rice, barley, durum wheat, and cropland pasture. All other crops were grouped into a category named others. They helped us put the crop-specific N fertilizer use rate, application timings, and application methods into a spatial context.

### 2.2.4 Crop phenology

We derived state-level crop phenology information from the USDA-NASS weekly crop progress report, which recorded the fractional acreage that has reached a given crop development stage (USDA-NASS, 2018). We linearly interpolated the weekly crop progress and identified the day at which crop development was 5%, 15%, 85%, and 95% completed. We extracted the planting and harvesting dates for all major crops except for cropland pasture. For winter wheat, we also obtained the date of dormancy breaking in the early spring (green-up) from 2014 to 2016. To gap-fill the planting date of a specific crop in a given state for missing years, we grouped states by latitude and adopted the distance-weighted interpolation (Eq. 3) using the mean date of the corresponding group.

$$Date_{i+k} = \frac{Mean_{i+k} \times Date_i}{Mean_i} \times \frac{k-i}{j-i} + \frac{Mean_{i+k} \times Date_j}{Mean_j} \times \frac{j-k}{j-i} \tag{3}$$

Where *Date* refers to the date of a given crop development stage that contains missing values, *Mean* refers to the mean date of the given stage of grouped states, the year *i* and *j* are the beginning and ending year of the gap, respectively, and *k* is the kth missing year.

The survey periods of crop progress provided by USDA-NASS vary across crops and states. For example, the data of durum wheat is available only in the years 2014 and 2015, while the data of barley started from 1996. The records of the other seven crops are available since the 1980s. To extend the crop-specific planting date records back to 1900, we adopted the approach used in the Environmental Policy Integrated Climate (EPIC) crop model (Williams et al., 1989), which considers daily heat unit accumulation (HU, Eq. 4) and heat unit index (HUI, Eq. 5) for crop phenological development estimation. It assumes that crops are ready to be planted or to break dormancy when the mean of daily maximum and minimum temperature equals to the base temperature (Tb) (i.e. when HU reaches 0), and to be harvested when the cumulative HU equals to potential heat units (PHU) (i.e. when HUI reaches 1). Based on the days at which 5%, 15%, 85%, and 95% crop development were completed between 1980-2015, we calculated the crop-specific Tb and PHU of each state with daily maximum and minimum temperature smoothed by a seven-day moving window from 1979 to 2015 for four percentages respectively. Instead of using the temperature at planting in fall as Tb, we used the temperature at green-up in early spring as Tb for winter wheat and fall barley to obtain a more accurate estimation of harvesting dates of these two crops. The averages of Tb and PHU in the earliest five available years of each crop type in each state were applied to Eq. 4 and Eq. 5 to calculate the dates of all four developments of all stages for the unavailable years back to 1900.

$$HU_k = \frac{Tmax_k + Tmin_k}{2} - Tb \quad HU_k > 0 \tag{4}$$

where *HU* is heat unit and indicates the planting date for spring-planted crops and green-up date for fall-planted crops when it reaches 0, *Tmax* and *Tmin* are daily maximum and minimum temperature in ℃, *Tb* is the crop-specific base temperature in ℃, *k* refers to the day k.

$$HUI_i = \frac{\sum_{k=1}^{i} HU_k}{PHU_j} \tag{5}$$

Where *HUI* is the heat unit index, which ranges from 0 to 1 and indicates the harvesting date when it reaches 1. *PHU* is the potential heat units required for harvesting, *i* and *k* are day i and day k, *j* refers to crop type j.

**2.2.5 Nitrogen fertilizer use dataset**

The historical state-level crop-specific N fertilizer use dataset (N fertilizer use rate, N fertilizer types, and application timing) of the U.S. were produced from our previous study (Cao et al., 2018), which includes N fertilizer use rate for 10 major crop types and others during the period 1900-2015.

We calculated the proportion of 11 major single N fertilizers in total fertilizer consumption in each state each year. They include Anhydrous Ammonia (AnA), Aqua Ammonia (AqA), Ammonium Nitrate (AN), Ammonium Sulfate (AS), Nitrogen Solution (NS), Sodium Nitrate (SN), Urea, Calcium Nitrate (CN), Diammonium Phosphate (DAP), Monoammonium Phosphate (MAP), and Ammonium Phosphates (APs). All other N fertilizers were grouped into others. We assumed there is no difference in the share of fertilizer types among crop types within the same state. Thus we split state-level crop-specific N fertilizer use into 12 N fertilizer categories according to this share ratio.

We allocated annual N fertilizer use generated above to daily application by considering N fertilizer application timing (USDA-ERS, 2015) and crop phenology information (USDA-NASS, 2018). According to the USDA survey, we calculated the ratio of four application timings to annual fertilizer consumption of each crop in each state. Four application timings are fall (previous harvest), spring (before planting), at planting, and after planting. Thus, we further split annual N fertilizer use into four application timings by each crop type each fertilizer type each state. We assumed that fall application occurs one month after harvesting, whereas before planting and after planting applications occur one month before and after planting date, respectively. Among the four dates of each crop phenological development stage (i.e., completion for 5%, 15%, 85%, and 95% area) we generated in section 2.2.4, the period between the dates of 15% and 85% completion is the most active range. Therefore, we assumed that 80% of N fertilizer use allocated in each application timing is applied to the active period (15%-85%), while the periods of 5%-15% and 85%-95% each receive 10%, respectively. We evenly spread the N fertilizer use over every day within the corresponding period of four application timings via dividing N fertilizer use rate by the number of days. After that, we aggregated the daily application to the monthly step. More details can be found in the example shown in Supplement Table S2 and Fig. S1. For winter wheat and fall barley, we allocated the use of N fertilizer after planting to the green-up stage in the following year. While for cropland pasture, we adopted the application timing strategy from Goebes et al. (2003), in which 1/30 of the total N fertilizer amount is applied in January, February, October, November, and December, 1/12 is applied in May, June, July, and August, and 1/6 is applied in March, April, and September. Because most crops in the others group such as oil seeds, legumes, small grains, fruits, and vegetables are spring-planted crops, we used the average ratio of monthly application to annual total over eight major crops by excluding winter wheat, cropland pasture, and fall barley in each state to extract the monthly application rate of all other crops.

USDA-ERS (2015) also reported how N fertilizer was applied and the percentage of acreage that was treated with the same method of each crop in the surveyed state. We regrouped the methods of the USDA survey according to the categories of the

REML model (Table S1). Specially, we assumed that broadcast and then flooded or incorporation and then flooded are only applied to rice. In addition, N fertilizer types AnA and AqA are only incorporated into the soil and NS is only applied as a solution. We calculated the planted area ratio of each application method of nine major crops of each state. We allocated N fertilizer use generated above to different application methods by using the area ratio. Thus, we generated monthly N fertilizer use rates under multiple application methods of each N fertilizer type of each crop type in each state.

Based on the U.S. gridded crop type distribution maps developed by Yu and Lu (2018), we assigned the aforementioned monthly crop-specific N fertilizer use rate of each N fertilizer type at each timing and by each application method into each 1 km × 1 km grid cell from 1900 to 2015. In addition, we converted the N fertilizer use rate from planting area-based (g N m$^{-2}$ cropland area per year) to grid area-based (g N m$^{-2}$ land area per year) by timing the spatialized N fertilizer use rate with the corresponding cropland density maps.

## 2.3 Factorial contribution assessment

Environmental factors and human activities have considerable impacts on the dynamics of NH$_3$ emissions. We set up five simulation experiments to quantify the roles of five major factors including temperature, cropland distribution, cropland rotation, N fertilizer type, and N fertilizer application rate, in shaping NH$_3$ emission since the 1960s (Table 1). The first simulation experiment (S1) was designed to mirror the temperature effect by keeping all other four factors unchanged at the level of 1960. We set up the rest simulation experiments (S2-S5) by adding the annual change of cropland distribution, cropland rotation, N fertilizer use rate, and N fertilizer type successively to S1. In S2, we allowed the percentage of cropland in each grid cell to change following the prescribed input data but kept the crop type within grid cells unchanged. Whereas in S3, the cropland percentage and type changed simultaneously through the study period. We further added annual N fertilizer use rates into S4 with N fertilizer type ratio fixed in 1960. We treated 1960 as the baseline year and run all the simulations from 1960 to 2015. The value difference between the simulated year and 1960 in S1 was calculated to estimate the temperature effect. We calculated the differences between S2 and S1, S3 and S2, S4 and S3, and S5 and S4 to assess the impacts of cropland distribution, cropland rotation, N fertilizer rate, and N fertilizer type, respectively.

## 2.3 Correlation between NH$_3$ emission and deposited NH$_4^+$ concentration

We obtained monthly site monitoring data of precipitation NH$_4^+$ concentration for the period 1985-2015 in the North America from the National Atmospheric Deposition Program (http://nadp.slh.wisc.edu/data/NTN/ntnAllsites.aspx). After aggregating the monthly data to spring (March-June) and full year at each site respectively, we generated the atmospheric NH$_4^+$ concentration maps using the Inverse Distance Weighting interpolation method and resampled the maps to 1 km resolution to make it comparable to our estimated NH$_3$ emission maps. The associations between fertilizer-induced NH$_3$ emission and NH$_4^+$ concentration in precipitation during 1985-2015 at each grid cell were examined using Pearson correlation coefficients with statistical significance at p < 0.01 and p < 0.001.

# 3 Results

## 3.1 Historical NH₃ volatilization from crops and N fertilizers

Our estimation indicates that the ratio of national $NH_3$ emission to total N fertilizer input declined from around 5.8% in the 1920s to below 4% in the 1970s, and then consistently rose back to 5.9% in the 2010s (Fig. 1a). We find that $NH_3$ emissions from synthetic N fertilizer in the U.S. remained less than 41 Gg N $yr^{-1}$ before 1950 and then sharply increased to 469 Gg N $yr^{-1}$ in 1981, followed by a slower rise to 641 Gg N $yr^{-1}$ by 2015 (Fig. 1a). Regionally, $NH_3$ emissions have consistently increased since the 1960s in the Northern Great Plains and the Northwest. Whereas the $NH_3$ emissions in the remaining regions have levelled off or slightly declined after peaking in the 1980s (Fig. 2).

Among all major crop types, $NH_3$ volatilized from corn accounted for over 40% of total fertilizer-derived $NH_3$ emission after 1960. Moreover, the increase in $NH_3$ emissions from corn fields was the major driver of the $NH_3$ emission growth in recent decades, contributing to 52% during 1980-2015, and 83% during 2000-2015 (Table S3). Although $NH_3$ emission from spring wheat accounted for less than 7% of total $NH_3$ emissions during 1980-2015, 14% of the fertilizer-$NH_3$ emission increase can be attributed to spring wheat production.

The contributions of N fertilizer types to total $NH_3$ volatilization varied in different periods (Fig. 1b). All other N fertilizer types, including single (i.e. Calcium ammonium nitrate) and mixed N fertilizers, are the dominant source of $NH_3$ emission before the 1960s (> 70%), during which the total $NH_3$ emissions were low. The contribution of urea-based fertilizers (Urea and Nitrogen solution) increased from 11.5% in 1960 to 68% in 2015, accounting for 78% of the fertilizer-induced $NH_3$ emission increase during this period (Fig. 1b, Table S4).

## 3.2 Spatiotemporal change in NH₃ volatilization

A large increase in $NH_3$ emissions was found across the U.S. from 1960 to 2015. Meanwhile, the hotspot of $NH_3$ emissions has shifted from the central U.S. to the Northern Great Plains and Minnesota (Fig. 3). Before 1960, most states in the US released less than 0.1 g $NH_3$-N $m^{-2}$ $yr^{-1}$, except the west and east coasts and a few states in the Midwestern U.S., such as Indiana, and Ohio (Fig. 3a). Since 1980, a tremendous increase of $NH_3$ volatilization (0.2-0.4 g N $m^{-2}$ $yr^{-1}$) occurred in the Midwest, the southern Great Plains, the Southeast, the Northwest, California, and Nebraska, with the highest $NH_3$ emission centered in Indiana and Ohio (0.4-0.6 g N $m^{-2}$ $yr^{-1}$) (Fig. 3b). $NH_3$ volatilization further enhanced after 2000, during which hotspots of $NH_3$ volatilization widely expanded in western Minnesota, Texas, and the western Southeast. (Fig. 3c). The most intensive $NH_3$ volatilization (> 0.6 g N $m^{-2}$ $yr^{-1}$) occurred in the northern Great Plains, the Northwest, and Minnesota in 2015 (Fig. 3d).

We find that the $NH_3$ loss proportion to total N fertilizer use remained less than 6% in the eastern U.S. before the 1980s. However, 6%-9% loss ratios are found in vast areas in the western U.S., with some areas in South Dakota, Nevada, and Utah losing up to 12% of N fertilizer via $NH_3$ (Fig. 4). After the 1990s, the Northern Great Plains, the Northwest, and part of the Southwest gradually became major players with an $NH_3$ loss proportion greater than 12%.

### 3.3 Monthly NH₃ emissions

Our results indicate that, in 2015, NH$_3$ emission levels were high in March, April, May, and June (Fig. 5). In addition, the emission hotspots showed large spatial variations over months (Fig. 5). Specifically, a vast amount of NH$_3$ (> 0.24 g N m$^{-2}$ month$^{-1}$) volatilized from the Midwest, the Northern Great Plains, and parts of the Northwest in April, while the southern North Great Plains and the eastern Midwest served as a major NH$_3$ source (> 0.24 g N m$^{-2}$ mont$^{h-1}$) in June. In contrast, NH$_3$ emissions in winter (Dec.-Feb.) and August were at a low level (< 0.04 g N m$^{-2}$ month$^{-1}$).

Monthly NH$_3$ emissions across the nation experienced a dramatic increase since 1960, especially during 1960-1980 (Fig. 6a). Meanwhile, NH$_3$ emissions in March and April showed large inter-annual fluctuations compare to other months. The NH$_3$ emissions in April have consistently increased by 64%, from 100 Gg N month$^{-1}$ in 1980 to 164 Gg N month$^{-1}$ in 2015, while the emissions in June and May slowly increased by 43% and 42%, from 70 Gg N month$^{-1}$ and 69 Gg N month$^{-1}$ to 100 Gg N month$^{-1}$ and 98 Gg N month$^{-1}$, respectively. NH$_3$ volatilized in April, May, and June together account for 70% of annual

emission (Fig. 6b). Before the 1960s, May dominated the annual emissions, followed by June and April. In this study, we find that maximum emissions have gradually shifted to earlier months and peaked in April since the 1960s. Interestingly, the reduction of emissions in June mainly occurred before the 1960s, whereas the rise of emissions in April mainly occurred after the 1970s (Fig. 6b).

Besides, our study indicates that the increment of April emission has widely distributed in the western U.S. since 1960, with

the largest increase (> 20%) found in the Great Plains and the Northwest (Fig. S2). On the contrary, although minor increases were found in the corn-belt, large decreases (< -10%) in May occurred in the Dakotas, Minnesota, and along the eastern coast of the U.S.

### 3.4 Factorial contributions

Our simulation experiments show that N fertilizer input is the dominant contributor to boost NH$_3$ emissions across the US

since 1960, especially in the Northeast, the Midwest, the Great Plains, and the Southwest (Fig. 7, Table S5). The roles of other factors affecting NH$_3$ emissions differed among regions and over periods. We find that temperature posed a weakly positive effect on NH$_3$ emission in most regions except the Northern Great Plains and Northwest during the simulation period. Cropland area and rotation changes overall led to decreases in NH$_3$ emission but had complicated impacts among regions since 1960. In the intensively managed regions, such as the Midwest, the Great Plains, cropland use change slightly

increased NH$_3$ emission, whereas decreased NH$_3$ emission in the Northeast and the Southwest regions (Fig. 7b, 7h). Crop rotation lowered NH$_3$ emissions in the US and most regions except the Northeast. Changes in N fertilizer type had largely increased NH$_3$ emission in regions such as the North Great Plains, the Northwest, and the Southeast, especially after the 1990s.

**4 Discussion**

 **4.1 Comparison with previous studies**

We compared our estimates of annual $NH_3$ emissions across the contiguous U.S. with the previously published results (Fig. 8). The magnitudes of $NH_3$ emission estimates differ significantly, ranging from 460 Gg N yr$^1$ to 756 Gg N yr$^{-1}$, among previous studies due to the difference of data sources (e.g. N fertilizer and cropland distribution) and estimation approaches (e.g. "bottom-up" and "top-down") they adopted. our estimated $NH_3$ emissions are much lower than those estimated by two

inventories, in which non-agricultural N fertilizer uses have been included (U.S. EPA, 2019) and the emission factor is less constrained by environmental drivers (Goebes et al., 2003). For example, our estimate of $NH_3$ emission in 1995 is 40% lower than an early study (i.e., 756 Gg N yr$^{-1}$ as estimated by Goebes et al. (2003) vs 504 Gg N yr$^{-1}$ in this study). This may be because the EF Goebes et al. (2003) used was only based on N fertilizer type, while we considered the combined effects of temperature, soil properties, crop type, and N fertilizer management to modify the EF. Whereas our estimated $NH_3$ emission

is very close to a "bottom-up" inventory (490 Gg N yr$^{-1}$ from Park et al. (2004) vs 521 Gg N yr$^{-1}$ from this study ) and a "top-down" estimation (540 Gg N yr$^{-1}$ from Gilliland et al. (2006) vs 512 Gg N yr$^{-1}$ from this study). In contrast, Two studies, considering more constrains on EF such as canopy absorption and wind speed, had smaller $NH_3$ estimations (Bash et al., 2013; Paulot et al., 2014). For example, The $NH_3$ emission of 2002 by Bash et al. (2013), which considered the effect of canopy absorption and release additionally, is 20% lower than our estimate (460 Gg N yr$^{-1}$ from Bash et al. (2013) vs 564 Gg N yr$^{-1}$

from this study).

We also compared the spatial pattern of $NH_3$ emissions in 2011 estimated by our study with that from the U.S. National Emissions Inventory (U.S. EPA, 2019). The spatial patterns revealed by these two studies were similar: the hotspots of $NH_3$ emissions concentrated in the Northern Great Plains, parts of the Northwest, the Corn-Belt, and the Rice-Belt, and relatively lower $NH_3$ emissions were found along the eastern coast of the U.S. (Fig. S3).

We further explored the monthly variations in the estimated $NH_3$ emissions among studies (Fig. 9). All studies agreed that the majority of $NH_3$ is released in spring and June while winter is the minimum $NH_3$ emissions season (Gilliland et al., 2006; Goebes et al., 2003; Pinder et al., 2006). April was commonly identified as the peak $NH_3$ emission month by three of these studies, which is consistent with our estimates. Whereas EPA-NEI, considering canopy uptake and release of $NH_3$, found a delayed peak in May. Several studies found relatively higher $NH_3$ emissions in March (Gilliland, 2003; Goebes et al., 2003;

Pinder et al., 2006). Our estimate in 2004 also showed the same pattern, in which the planting date was early. However, $NH_3$ emission in March was very low in 2011 from our study due to the delayed planting date and thus resulted in greater April $NH_3$ emission. Owing to the limited data of actual fertilizer use history, these studies used the recommendation from fertilizer experts to spread the N fertilizer after planting over February and March in the following year at the green-up of winter wheat with a fixed ratio. However, we allocated this proportion of N fertilizer to early spring based on the annual green-up date of

winter wheat in each state derived from USDA-NASS. The discrepancies in crop phenology and N fertilizer application timing also introduced more disagreements to the secondary peak in fall (Fig. 9). Different from a single large peak in October

estimated by EPA-NEI, other studies found two smaller peaks in September and November. Our results, however, indicated relatively smaller emissions compared to other studies, which is because the ratio of N fertilizer application in fall we extracted from USDA-ERS is smaller. Although some states, such as Iowa, applied a considerable proportion of annual N fertilizer input in fall, the N fertilizer use in fall is low in the entire US. In summary, our estimate of monthly $NH_3$ emission is generally consistent with other studies but with large inter-annual variations based on crop phenology survey.

## 4.2 Spatiotemporal change in the $NH_3$ emissions

The "V" shape of historical national and regional $NH_3$ emission factors mainly resulted from the changing preference in using different N fertilizer types (Cao et al., 2018). The decline in the early stage from the 1920s to the 1970s was due to the decrease in the use of Ammonium sulfate, while the rising emission factor from the 1970s to the present was caused by the popularity of Urea-based fertilizer (Fig. 1b, Fig. 7, Table S4). In addition, the N fertilizer use hotspots shifting to more alkaline areas, such as the Northern Great Plains may contribute to this increasing trend. The $NH_3$ emission factor estimated by our study is close to 6% in the U.S. and is significantly lower than the estimated global EFs, ranging from 11% to 14% (Bouwman et al., 2002; Paulot et al., 2014; Vira et al., 2019). This indicates that agricultural management in the U.S. is more efficient in reducing $NH_3$ loss compared to other counties. However, the $NH_3$ emission factor varied substantially across the U.S., ranging from 2.5% to 29% (Fig. 4). We found the highest loss proportion (> 12%) in the Northern Great Plains and the Northwest. Adopting better N fertilizer management practices, such as appropriate application timing and method, is recommended to reduce $NH_3$ emission in these high loss regions.

$NH_3$ emissions from synthetic N fertilizer in the U.S. increased rapidly during 1960-1980, which may be attributed to cropland expansion (Nickerson et al., 2011) and the dramatic increase in N fertilizer use rate in most crop types (Cao et al., 2018). However, the national increases in total $NH_3$ emissions from fertilizer use slowed down after 1980. Compared to the stable or declining trend in the other five regions, the $NH_3$ emission of Northern Great Plains and the Northwest kept increasing to recent years, which contributed to the post-1980 increase of national $NH_3$ emissions (Fig. 2). We recognized $NH_3$ emissions from corn and spring wheat dominated the increase in total $NH_3$ emissions after the 1980s (Fig 1a and Table S1). The conclusion drawn from our factorial contribution analysis shows that changes in cropland area and rotation have a minor influence on $NH_3$ emission in the nation (Fig. 7), which is primarily because N fertilizer input was kept constant at the level of 1960. Besides, the cropland area changes represent the summation of cropland expansion and abandonment across the country, resulting in a relatively small contribution to $NH_3$ emission increases. USDA Crop Production Historical Report shows that the largest increases in planted areas among all non-legume crop types from the period of 1960-1980 to the period of 1995-2015 were corn and spring wheat, increased by 12% and 22% respectively (Fig. S4). Specifically, the increases in corn and spring wheat planting area were mainly found in Kansas, Minnesota, and the states in the northern Great Plains and the Northwest (Fig. S4). In addition, the average N fertilizer use rates of corn and spring wheat have grown to be the second and third highest among other crop types since 2000 (Cao et al., 2018). Therefore, the rapid increase of corn and spring wheat

cropland area combined with high N fertilizer use rate in the Northern Great Plains and the Northwest contributed to the increasing U.S. $NH_3$ emissions after the 1980s (Cao et al., 2018; Nickerson et al., 2011; Yu and Lu, 2018).

Urea-based fertilizer has been proven to trigger high $NH_3$ volatilization (Sommer et al., 2004). With two major urea-based fertilizer types, Urea and Nitrogen Solution, increased by over 4000% and 300% since 1980, respectively (Fig. S5), the northern Great Plains and the Northwest have grown to be the most urea-based fertilizer used regions since 1980 (Cao et al., 2018), which contributed to the steep increase of $NH_3$ emission during this period (Fig. 2, Fig. 7). Even worse, the alkaline soil in the Northern Great Plains and the Northwest led to a high risk of $NH_3$ emission compared to other regions. For example, Iowa and Illinois in the Midwest received the most intensive N fertilizer in 2015 (Cao et al., 2018) but they emitted less intensive $NH_3$, which might be due to the neutral to weak acidity soil. Under the enhanced effect of alkaline soil in the Northern Great Plains and the Northwest, the increasing urea-based N fertilizer use and the northwestward corn and spring wheat expansion together greatly boosted the $NH_3$ loss proportion, which may contribute to the decreasing crop NUE in these regions (Lu et al., 2019) (Fig. 4). Although soil acidification through long-term agricultural land use may offset the effects of the increasing use of urea-based fertilizer, more effective policies and agricultural management are still needed in those high $NH_3$ loss proportion regions (Veenstra and Lee, 2015; Dai et al., 2018), which can prevent air quality deterioration and enhance crop NUE. Applying urease inhibitor with urea-based fertilizer was proved an effective practice to decrease $NH_3$ loss (Pan et al., 2016; Soares et al., 2012; Tian et al., 2015). In addition, 4R management (Right fertilizer source, Right rate, Right timing, and Right place) is effective in mitigating high $NH_3$ emissions.

### 4.3 Monthly peak of $NH_3$ emissions shifting from 1930 to 2015

The application timings of N fertilizer differ dramatically across the U.S. (Cao et al., 2018), which highly influence the seasonality of $NH_3$ emissions in different regions (Paulot et al., 2014). Corn and spring wheat producers in the Midwest, the Northern Great Plains, and the Northwest apply most of N fertilizer in spring before planting (Cao et al., 2018), resulting in the largest $NH_3$ emission in April (Fig. 2). Whereas farmers in the Southern Great Plains prefer to apply most of N fertilizer after planting for cotton and a considerable amount of N fertilizer at green-up for winter wheat, resulting in peaks in summer and early spring. As corn and spring wheat expanded into Minnesota, the Northern Great Plains, and the Northwest, as well as the increased use in urea-based fertilizer, $NH_3$ emissions from these areas rapidly gained the weight of total $NH_3$ emissions of the country. The hotpots of $NH_3$ emission shifted from the central US to the Northern Great Plains and Minnesota (Fig. 3). This change advanced the monthly $NH_3$ emission peak at the national scale (Fig. 6). In addition, the monthly peak shifting was also driven by an advanced crop planting date. Due to the development of genotypes and improvement of agricultural management and equipment, the corn-planting date became earlier by approximately two weeks between the 1980s and the 2000s in the corn-belt (Kucharik, 2006). In addition, widespread springtime warming across much of North America has also pushed toward an earlier planting date since the 1940s (Duvick, 1989; Hu et al., 2005; Schwartz et al., 2006).

## 4.4 Effects of increasing $NH_3$ emissions on $NH_4^+$ deposition

Although the central U.S. is the hotspot of $NH_4^+$ deposition, the largest increase in wet $NH_4^+$ deposition was found in the northern Great Plains and Minnesota from 1985 to 2015 (Du et al., 2014; Li et al., 2016). Our result shows that the increase of $NH_3$ emissions from synthetic N fertilizer in the Northern Great Plains, the Northwest, and Kansas was significantly correlated to the increase of $NH_4^+$ concentration in precipitation during 1985-2015 (Fig. 10, Fig. S6). $NH_4^+$ deposition is highly affected by local $NH_3$ emissions because $NH_3$ volatilized into the atmosphere has a very short lifetime and deposits close to the source quickly. Therefore, In addition to growing forest fire and livestock numbers (Abatzoglou and Williams, 2016), our study reveals that $NH_3$ emissions from increasing N fertilizer use played an important role influencing the inter-annual variability of $NH_4^+$ deposition in the northwestern U.S. over recent decades. Whereas with decreasing $NH_3$ emissions from N fertilizer in parts of Washington, Wisconsin, Florida, the Southeast and the Northeast since 1980 (Fig. 2), the $NH_4^+$ deposition promoted by an increasing forest fire, rapid urbanization, and growing livestock population (Fenn et al., 2018) showed strong negative relations with $NH_3$ emissions in these regions. In addition to $NH_4^+$ deposition, the PM2.5 also showed an increasing trend in Minnesota, the Northern Great Plains, and the Northwest from 2002 to 2013 (U.S. EPA, 2019). Since $NH_3$ in the atmosphere heavily involves in formatting $PM_{2.5}$, the increase of fertilizer-induced $NH_3$ emissions may contribute to the $PM_{2.5}$ increase in these regions. Therefore, the increase of $NH_3$ emissions induced by northwestward corn and spring wheat expansion and consequent urea-based fertilizer use might largely enhance the environmental stress in these regions.

## 4.5 Uncertainty

The major uncertainty sources in this study include the following aspects. (1) state-level N management data (rate, application timing, application method, and the fraction of each N fertilizer type) were used to calculate $NH_3$ emissions over the contiguous U.S. in our study because of the paucity of sub-state details. (2) The crop-specific N Application timing and method derived from the latest survey years were assumed to be unchanged over time due to the scarcity of inter-annual survey data. This assumption may cause bias in the monthly pattern of $NH_3$ emissions. For example, urea-based fertilizer, which is suitable for spring application, has been increasingly used to replace fall-applied anhydrous ammonia since 1960, we may underestimate $NH_3$ emissions in fall before 2000. (3) The ratio of each N fertilizer type was assumed to be constant across crop types in each state in a year. This may cause biases because farmers may apply different types of N fertilizers to different crops. (4) We allocated each N fertilizer type to the same application timing for each crop of each state based on state-level crop-specific application timing. However, farmers may only apply a certain N fertilizer at the time when its maximum profit can be achieved. For example, due to the high potential loss, Nitrogen Solution is seldom applied in fall after harvest. (5) Although we considered the effects of temperature, soil properties, crop type, and N fertilizer type and application method on $NH_3$ emission estimate, other factors such as wind speed, soil moisture, nitrification and urease inhibitors, and different N fertilizer use rate level may also significantly influence $NH_3$ emissions (Behera et al., 2013; Jiang et al., 2017; Lam et al., 2017). Increasing evidence suggests that $NH_3$ emissions increase exponentially with increasing N fertilizer rate (Jiang et al., 2017).

Zhou et al (2015) developed a nonlinear Bayesian tree regression model as a function of N fertilizer rate to estimate $NH_3$ emission in China and found the estimates match well with observations and satellite-based products. Thus, we may underestimate $NH_3$ emissions under a high N fertilizer use rate. Another example is the use of nitrification and urease inhibitors. Nitrification inhibitors have been found to increase $NH_3$ loss while urease inhibitors can limit $NH_3$ volatilization (Lam et al., 2017). Therefore, the uncertainty of usage of nitrification and urease inhibitor is likely to misrepresent $NH_3$ emissions. In addition, considering the bidirectional exchange process may improve the accuracy of seasonal $NH_3$ emission estimation (Bash et al., 2013). However, our work builds upon the newly-developed N fertilizer management and crop phenology dataset that combines crop-specific N fertilizer use rate, fertilizer type, application timing, application method, and phenology for each state ranging from 1900 to 2015. The REML model we are using makes sufficiently use of these information and provides higher levels of details over space and time.

## 5 Conclusion

This study comprehensively examined the spatiotemporal patterns of $NH_3$ emission owing to the changes in temperature, cropland area, rotation, and N fertilizer management in the U.S. from 1900 to 2015. We also examined the relationship between $NH_3$ emission and $NH_4^+$ concentration in precipitation over the last three decades. The gridded monthly time-series estimations of $NH_3$ emission, at a spatial resolution of 1 km $\times$ 1km, could serve as a solid database for national and regional air quality modeling and N budget assessment.

Our results indicate that $NH_3$ emission from synthetic N fertilizer uses in the U.S. rapidly increased from $< 50$ Gg N yr$^{-1}$ before the 1960s to 641 Gg N yr$^{-1}$ in 2015, among which corn and spring wheat are the major contributors. In addition, enhanced use of urea-based fertilizers strengthened the N loss through $NH_3$ emission after 1960. Spatially, the intensive $NH_3$ emission spots have shifted from the central U.S. to the northwestern U.S. since 1960 due to the northwestward cropland expansion onto the alkaline soils. Springtime warming, cropland expansion, and N fertilizer management practices change also altered the seasonal pattern of $NH_3$ emission in the U.S., shifting the peak emission month from May to April since the 1930s. Moreover, our analyses reveal that the increasing deposited $NH_4^+$ concentration in the Northern Great Plains could be greatly attributed to the increasing $NH_3$ emission in this region since 1985. In summary, our work highlights the importance of comprehensively assessing the environmental consequences of agricultural production. We call for proper fertilizer management practices in reducing $NH_3$ emission and improving nitrogen use efficiency.

**Data availability.** The estimated $NH_3$ emission dataset derived from this study is publicly available via figshare at https://doi.org/10.6084/m9.figshare.11692038.v3 (Cao et al., 2020).

**Author contributions.** CL and PC designed the research. PC compiled the database and carried out the data analysis. PC drafted the manuscript, with the guidance of CL and JZ. All the co-authors contributed to and reviewed the manuscript.

**Competing interests.** The authors declare that they have no conflict of interest.

**Acknowledgements.** This work was supported by the Iowa Nutrient Research Center, and NSF Dynamics of Coupled Natural and Human Systems (CNH) program (1924178).

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

Table 1. Experiments designed in this study

| Experiments | Abbr | Tem | Distribution | Rotation | Nfer rate | Nfer type |
|---|---|---|---|---|---|---|
| Tem only | S1 | 1960-2015 | 1960 | 1960 | 1960 | 1960 |
| Tem + Dis | S2 | 1960-2015 | 1960-2015 | 1960 | 1960 | 1960 |
| Tem + Dis + Rot | S3 | 1960-2015 | 1960-2015 | 1960-2015 | 1960 | 1960 |
| Tem + Dis + Rot + Nfer rate | S4 | 1960-2015 | 1960-2015 | 1960-2015 | 1960-2015 | 1960 |
| Tem + Dis + Rot + Nfer rate + Nfer type | S5 | 1960-2015 | 1960-2015 | 1960-2015 | 1960-2015 | 1960-2015 |

Note: Tem refers to temperature, Dis refers to cropland distribution, Rot refers to Rotation, Nfer rate refers to N fertilizer use rate, Nfer type refers to N fertilizer type. S5 allows all the factors to change through the study period, and provides the major results of this study


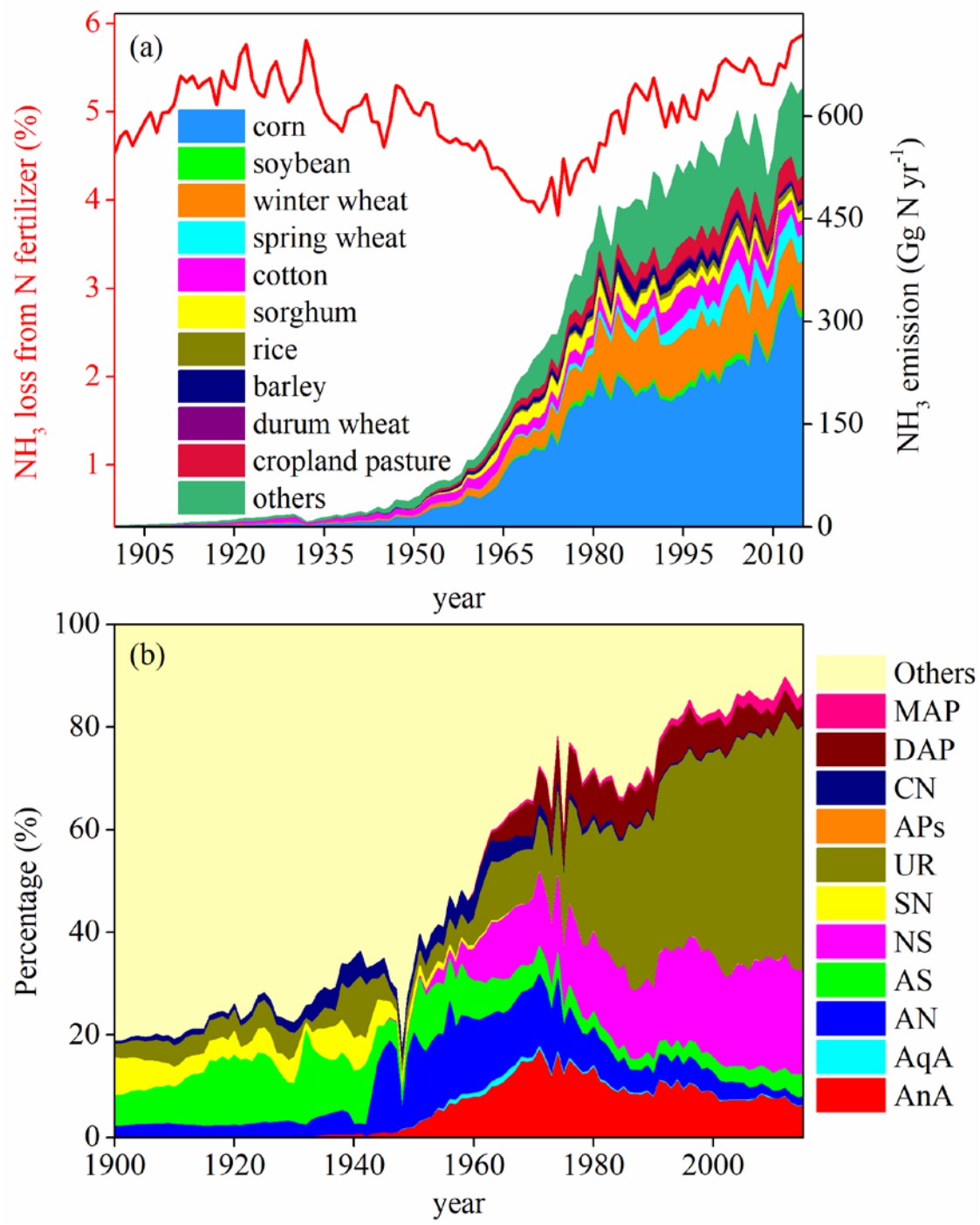

**Figure 1. Contributions of major crop types and N fertilizer types to historical NH₃ emissions since 1900. (a) Crop specific NH₃ emissions, (b) Relative contributions of 12 major N fertilizer types to annual total NH₃ emission. Solid line in (a) refers to the NH₃ loss percentage to total N fertilizer input.**


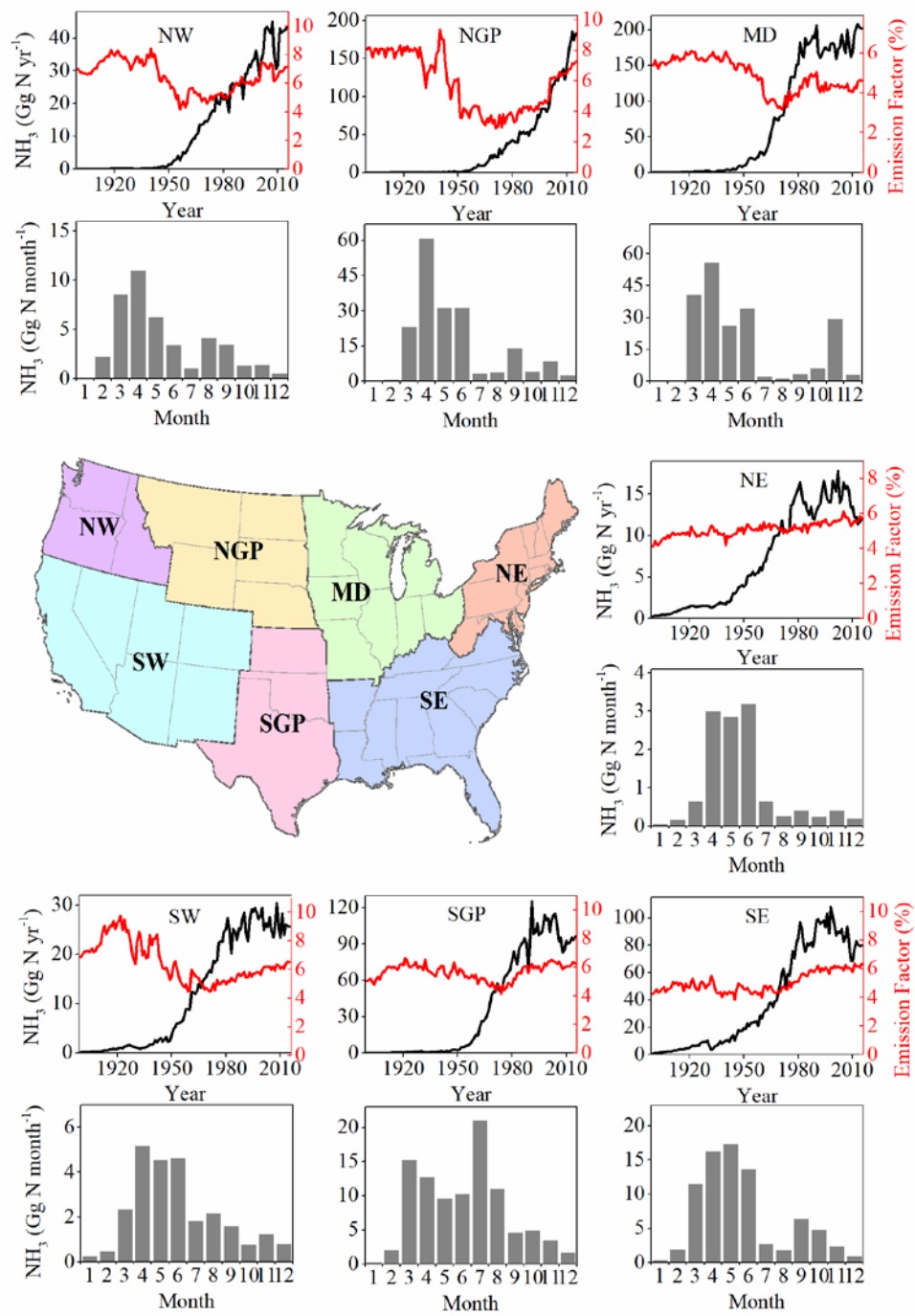

**Figure 2. Temporal and monthly pattern of NH₃ emissions in seven regions of the United States. The seven regions include the Northwest (NW), the Northern Great Plains (NGP), the Midwest (MD), the Northeast (NE), the Southwest (SW), the Southern Great Plains (SGP), and the Southeast (SE). Annual NH₃ emission is shown by black lines. Red lines represent the proportion of NH₃ emission to N fertilizer input. Gray bars indicate monthly NH₃ emissions of each region in 2015.**



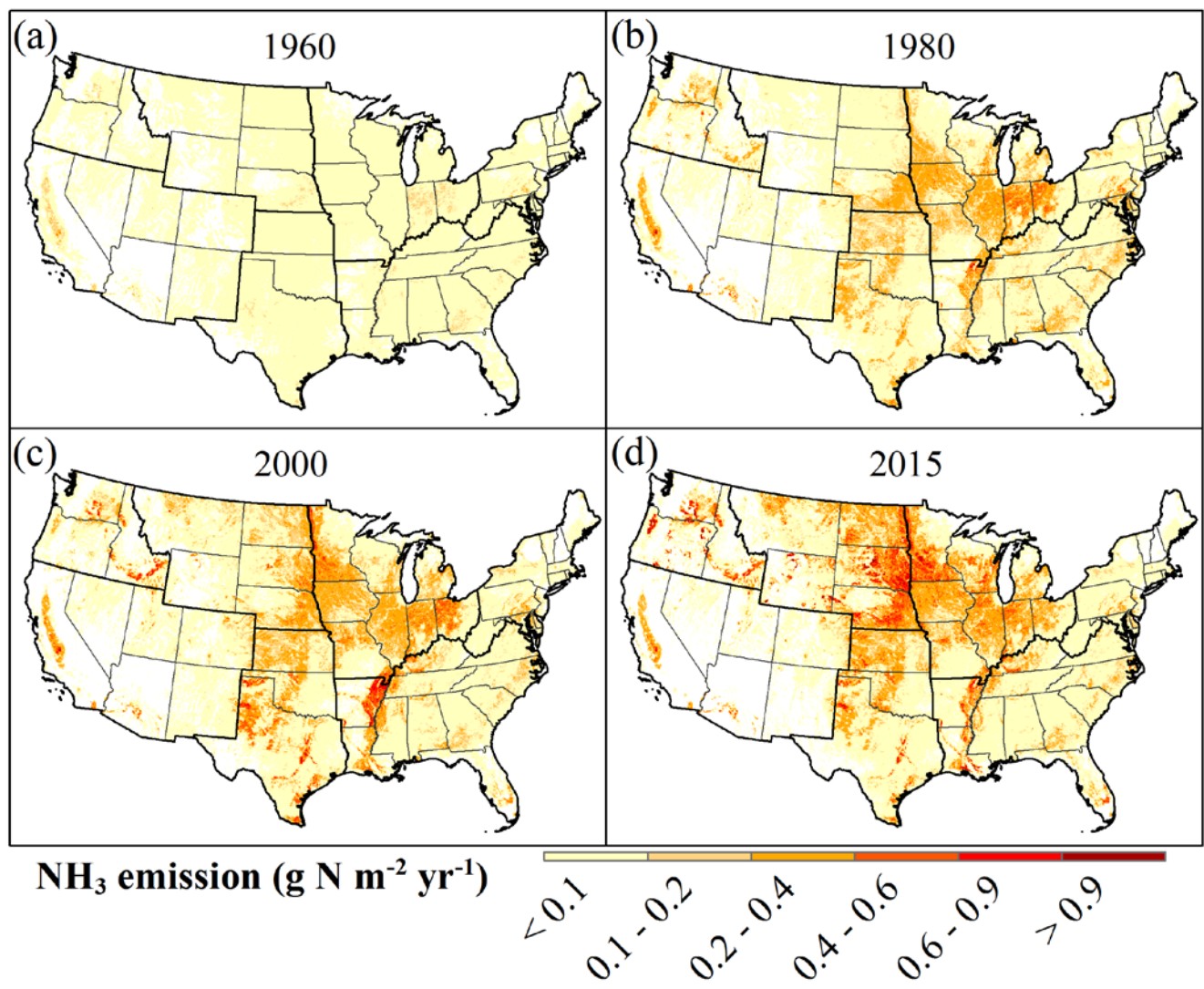

**Figure 3. Spatial distribution of NH₃ emissions in the U.S. from 1960 to 2015. Values represent NH₃ emission from synthetic N fertilizer applied over all crops in each 5 km by 5 km grid cell.**

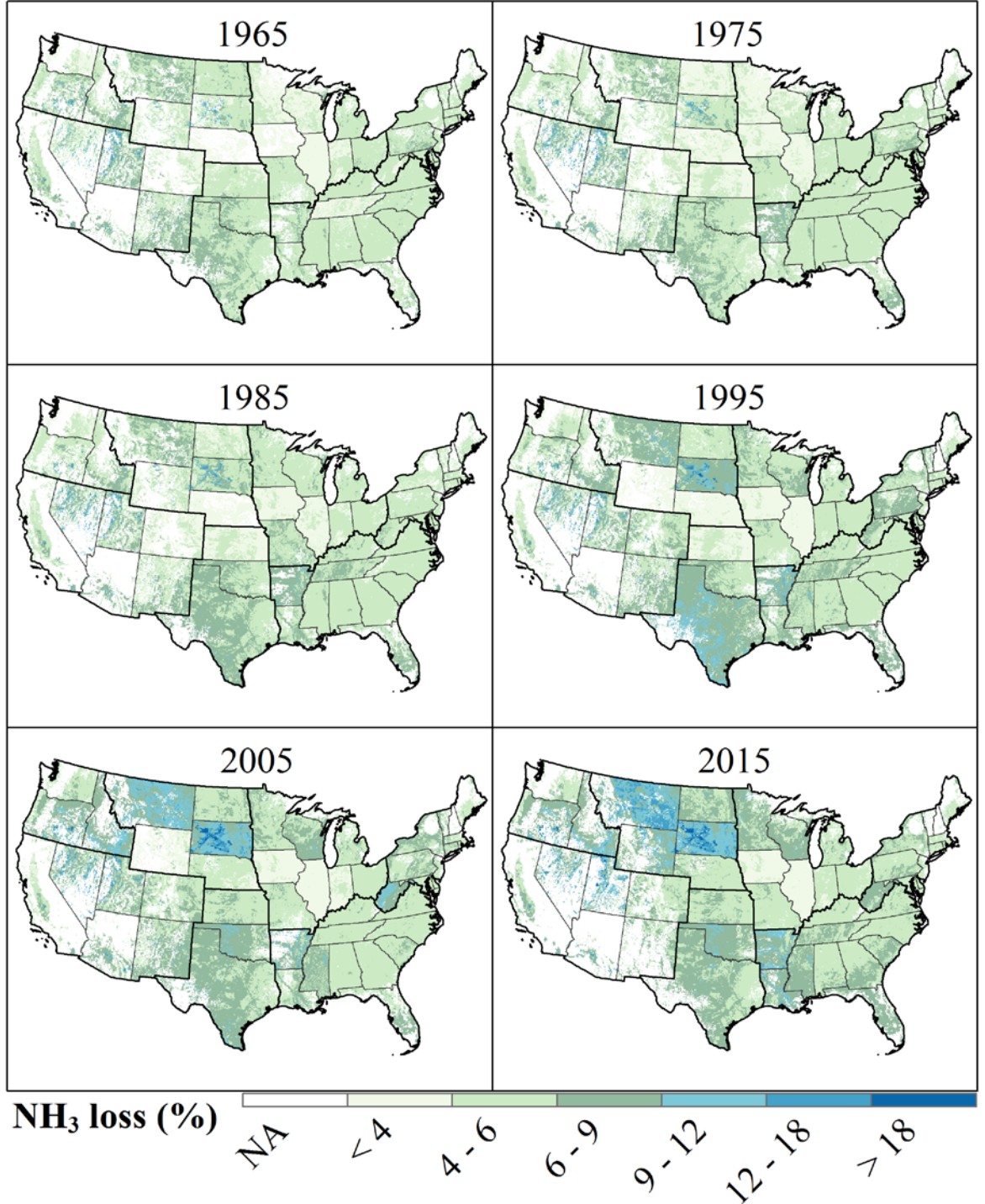


**Figure 4. Spatial and temporal patterns in NH₃ loss proportion relative to total N fertilizer input in the U.S. (the middle year of each decade is selected as an example).**

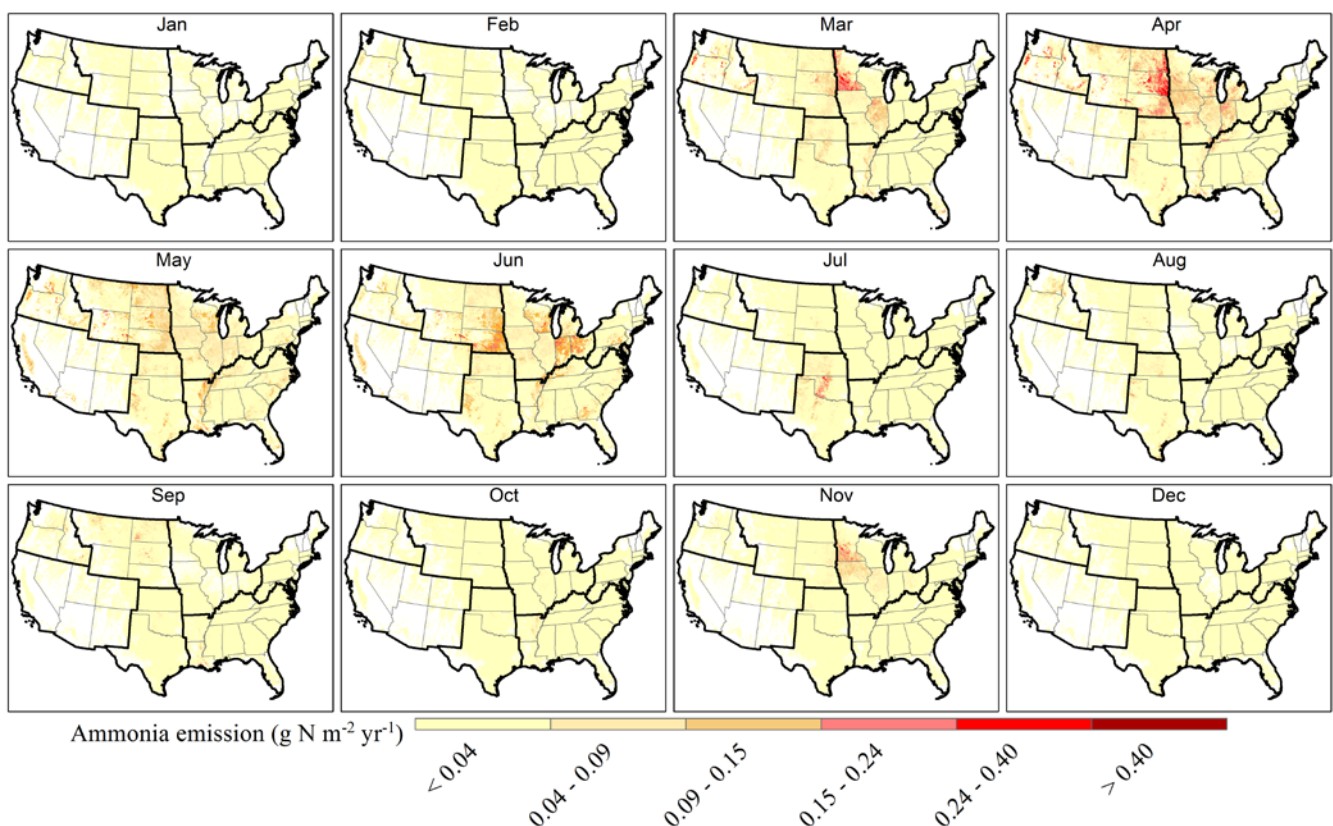

**Figure 5. Spatial distribution of monthly estimated NH₃ emission across the U.S in 2015.**

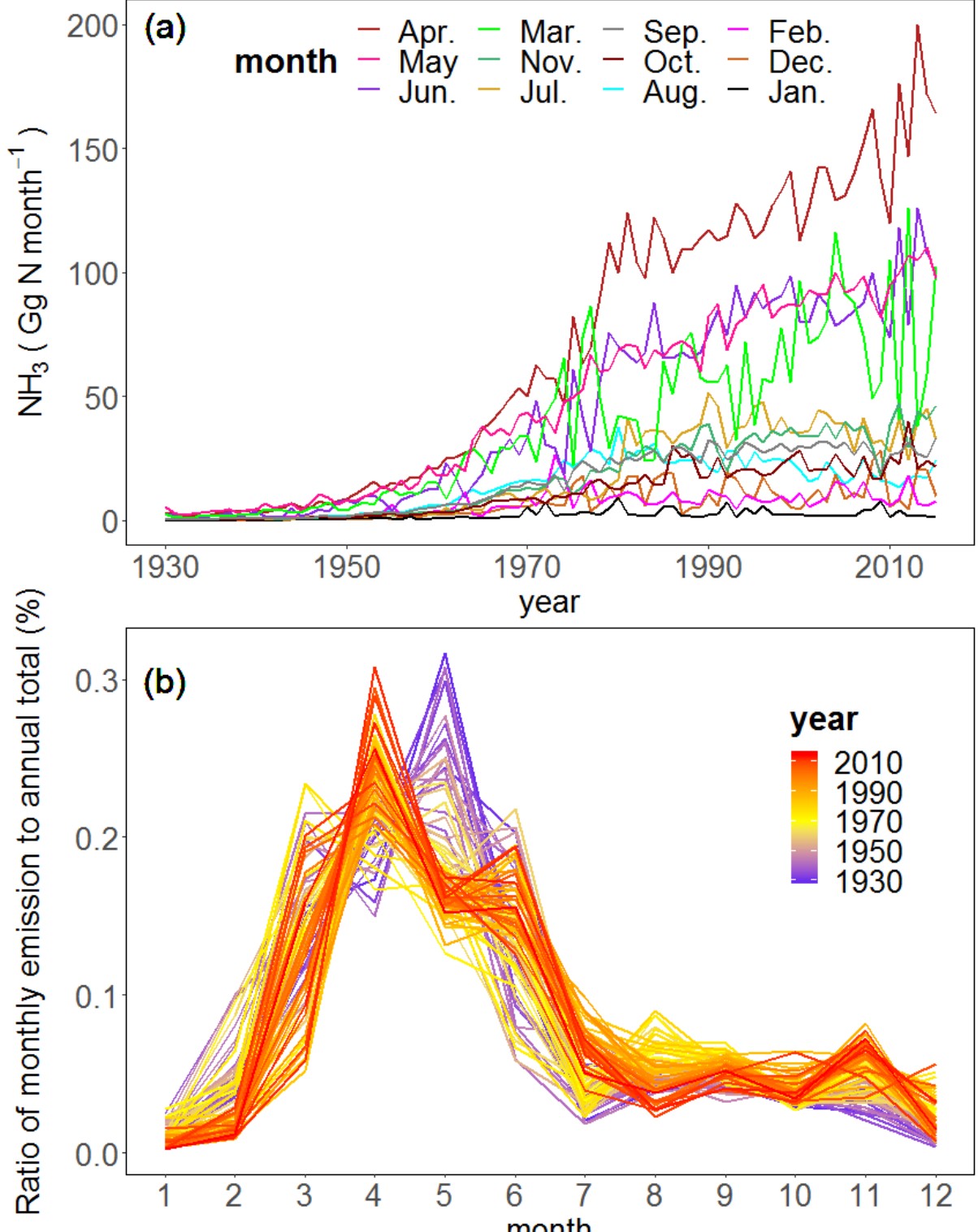

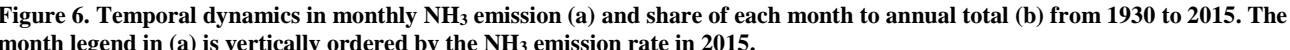

**Figure 6. Temporal dynamics in monthly NH₃ emission (a) and share of each month to annual total (b) from 1930 to 2015. The month legend in (a) is vertically ordered by the NH₃ emission rate in 2015.**

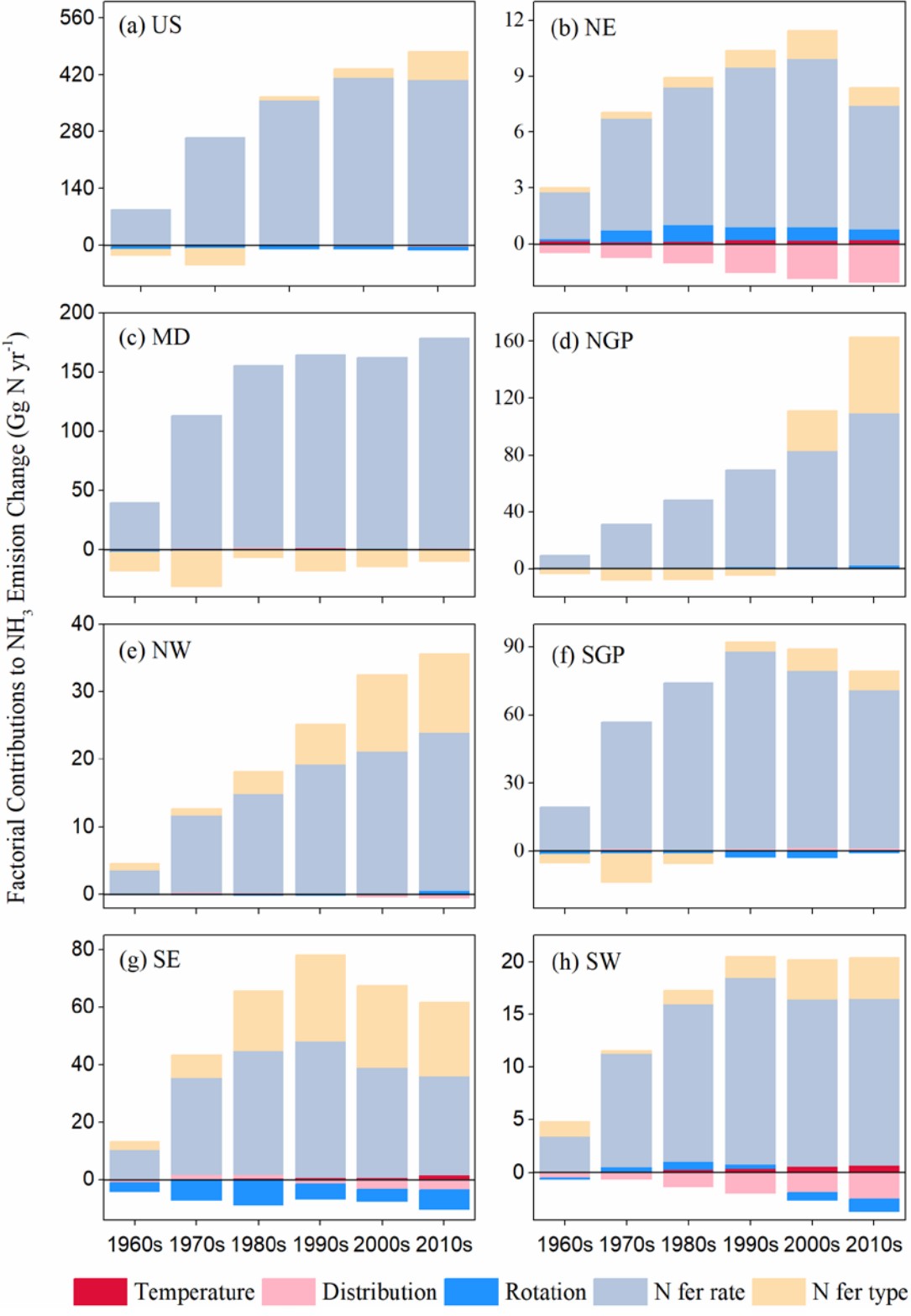

Factorial Contributions to NH₃ Emission Change (Gg N yr⁻¹)

Legend: Temperature | Distribution | Rotation | N fer rate | N fer type

**Figure 7. Decadal-average factorial contributions of temperature, cropland distribution, cropland rotation, N fertilizer use rate, and N fertilizer type to NH₃ emission change in the contiguous US and seven sub-regions. The seven sub-regions are the Northeast (NE), the Midwest (MD), the Northern Great Plains (NGP), the Northwest (NW), the Southern Great Plains (SGP), the Southeast (SE), and the Southwest (SW).**

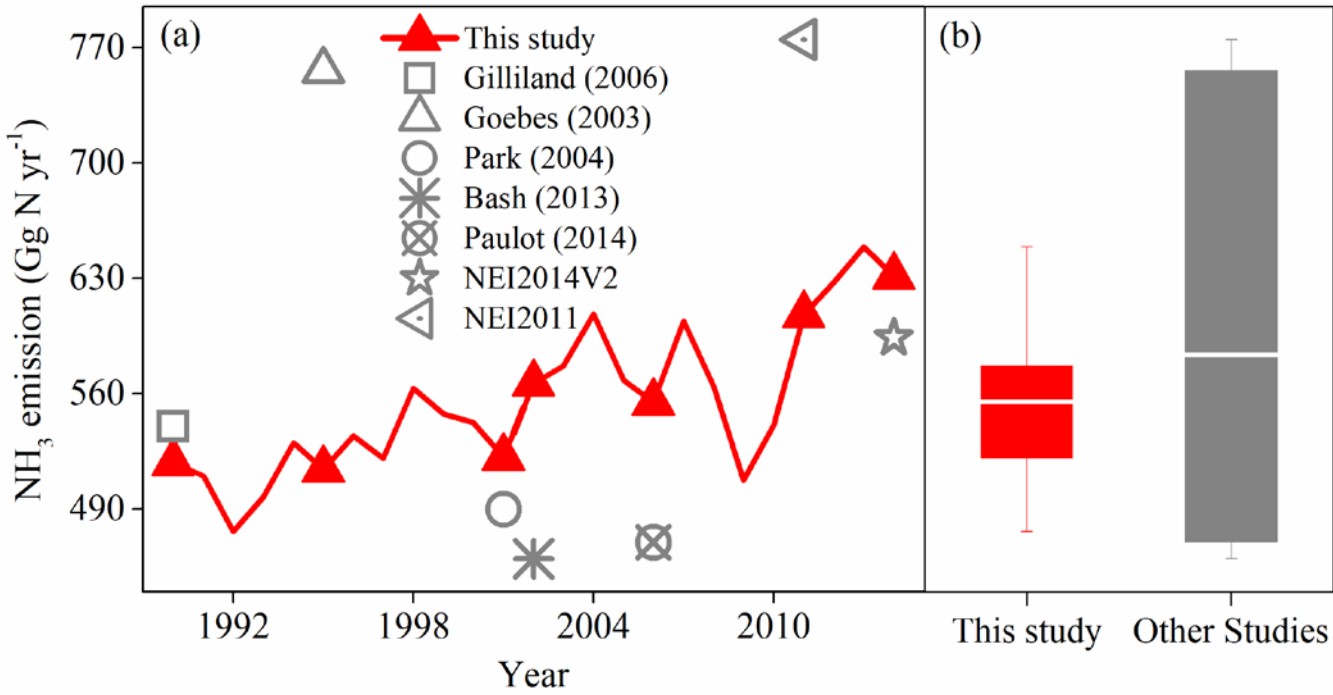

**Figure 8. Comparison of annual NH₃ emission estimates between this study and others. (a) Paired comparison between our result and individual research, (b) Boxes include 25-75% of NH₃ emission of all chosen years estimated by our study and other studies, respectively. White lines are mean values, and whiskers represent the min-max range of data. NH₃ emission estimated by Paulot et al. (2014) represents the average of 2005-2008, against which we compared our estimate of 2006.**


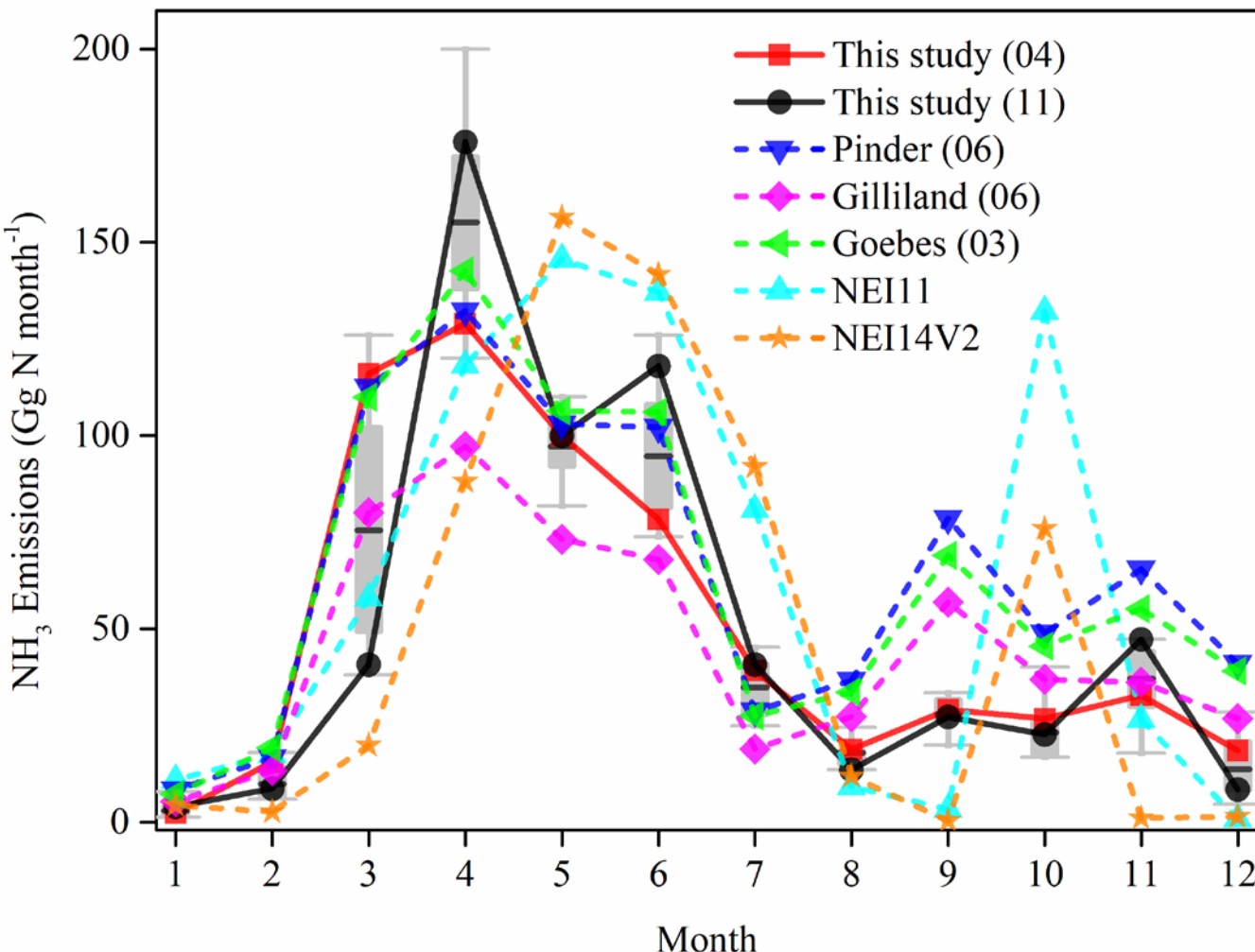

**Figure 9. Comparison of monthly NH₃ emission patterns between our estimate and other studies. Two typical monthly patters of NH₃ emission in this study were used. The estimate of 2004 represents the pattern when planting date is early, whereas the simulation of 2011 stands for the pattern when planting date is delayed. Two simulations using different approaches by EPA-NEI were chosen in the comparison. Grey boxes include 25-75% of monthly NH₃ emissions during 2005-2015 estimated by this study black lines are mean values, and whiskers comprise the whole range of data.**

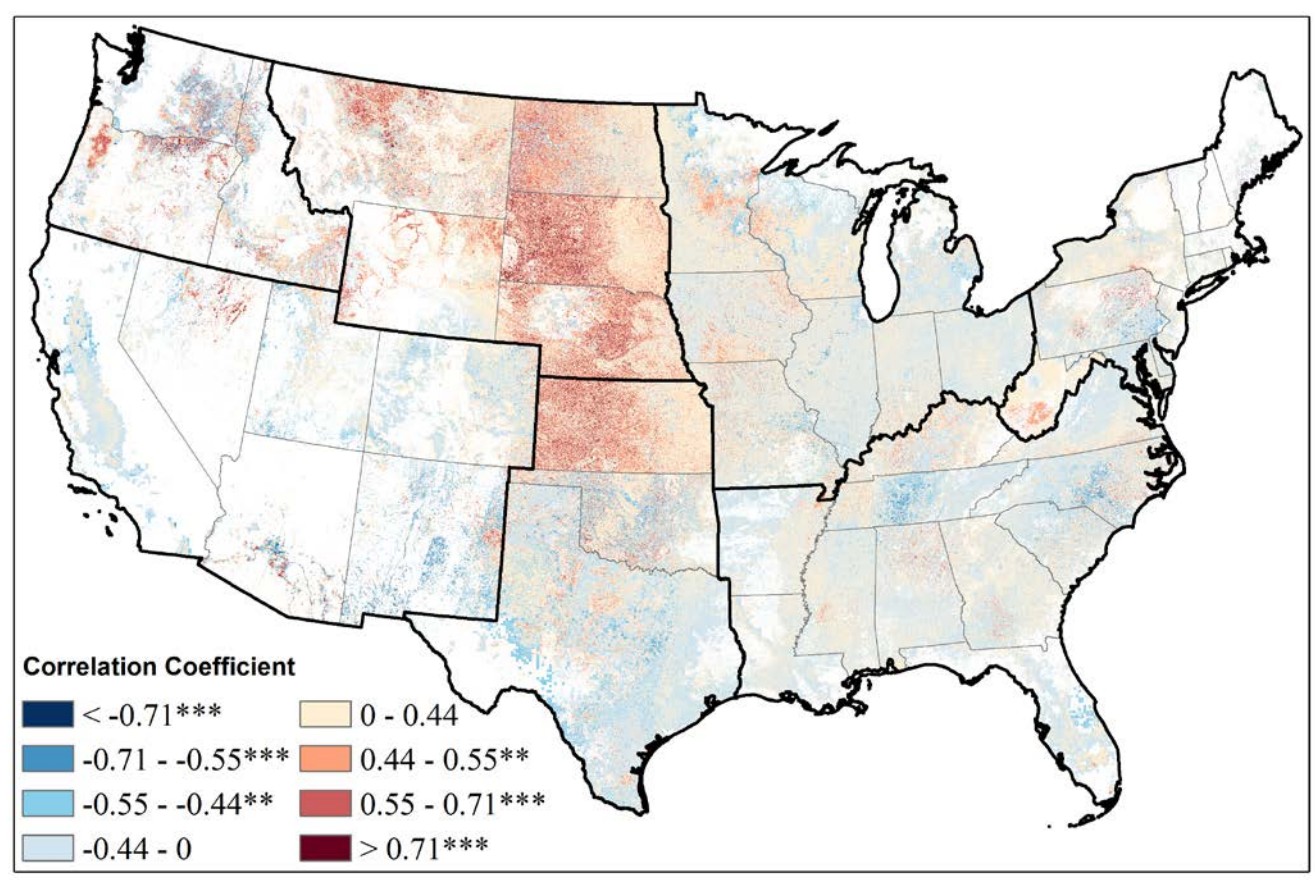

**Figure 10. Correlation coefficient between annual NH₃ emission from N fertilizer uses and annual NH₄⁺ concentration in precipitation between 1985 and 2015. The correlation coefficient was calculated between the two time series at each 1km × 1km grid cell. ** refers to P-value < 0.01, *** refers to P-value < 0.001.**