# Peer review of "Northwestward Cropland Expansion and Growing Urea-Based Fertilizer Use Enhanced NH3 Emission Loss in the Contiguous United States"

_Atmospheric Chemistry and Physics, 2020_

## Referee Comment (RC1) · Anonymous Referee #1 · 21 Apr 2020

In this study, Cao et al. derive US NH3 emissions associated with fertilizer application from 1900 to 2015. The strength of this study lies in the use of spatially-explicit time-series for cropland distribution and fertilizer application. The authors rely on a very simple emission scheme to estimate NH3 emissions. While this is acceptable considering the goal of this study, better quantification of the role of each factors and associated uncertainties for the authors' conclusions are needed before publication can be considered.

General comments

line 130 How would application of fertilizer at emergence (early spring) for winter wheat

impact the authors' conclusions

line 305 relationship with wet deposition is not very compelling. As noted by the authors there are a lot of different factors that could be at play. I would suggest to focus on spring and fall months where the authors expect the fertilizer contribution to be maximum

Trend attribution —————

I recommend the authors better quantify the relative importance of the different factors that contribute to changes in the magnitude and seasonality of NH3 emissions. I would suggest the authors perform their analysis using a climatology for a) temperature, b) fertilizer type, c) spatial crop distribution, e) crop mix

There are two important factors that I would like the authors to analyze in more details a) planting dates The authors rely on a climatology for planting dates. However, Kucharik (2006) showed using the USDA crop report that corn planting took place ~2 weeks earlier in 2005 relative to 1980. This dataset is available for other crops and it would be useful for authors to assess the impact of changing planting dates over this time period.

There also exists simple parameterizations to estimate planting dates based on temperature/precipitation that I would recommend the authors consider to estimate the variability in planting dates before 1979 (e.g., Bondeau (2007))

b) could the authors comment on the impact of long-term acidification that has been reported in several studies

Veenstra, J.J. and Lee Burras, C. (2015), Soil Profile Transformation after 50 Years of Agricultural Land Use. Soil Science Society of America Journal, 79: 1154-1162. doi:10.2136/sssaj2015.01.0027 Fuqiang Dai, Zhiqiang Lv, Gangcai Liu. (2018) Assessing Soil Quality for Sustainable Cropland Management Based on Factor Analysis and Fuzzy Sets: A Case Study in the Lhasa River Valley, Tibetan Plateau. Sustainability 10:10, pages 3477

Comparison with other inventories ——————

the authors need to compare their inventory against other efforts to develop historical emissions from EPA, EDGAR, and CMIP6. I believe that only gridded NH3 emissions from agriculture may be readily available from EPA and CMIP6 but I encourage the authors to contact the inventories' developers to obtain their estimates for historical US fertilizer emissions.

http://www.globalchange.umd.edu/ceds/ -> code is freely available
https://edgar.jrc.ec.europa.eu/

——

Technical comments:

line 30: please rephrase to more clearly separate the impacts associated with N deposition and with PM2.5

line 70 I would recommend discussing alternative (more recent) approaches used to derive NH3 emissions not only in the US but also in China and Europe. There have been a lot of progress in NH3 inventories since the work of Bouwman and the authors need to better explain why this approach was selected.

line 42 grammar: for quantifying long-term spatially explicit of NH3 emissions

line 63 objects -> goals

Line 136 The authors need to clarify that this dataset represents a climatology of present-day planting dates.

line 196 I am not sure what reportedly means in this context

---

## Referee Comment (RC2) · Anonymous Referee #1 · 21 Apr 2020

I forgot to mention this recent study that the authors also need to consider:

https://onlinelibrary.wiley.com/doi/pdf/10.1111/gcb.14499

---

## Referee Comment (RC3) · Anonymous Referee #2 · 26 Apr 2020

The manuscript by Cao et al. estimates NH3 emissions from fertilizer in the US over the past century. By tracking different types of fertilizer and crops, they identify variability in the spatial distribution of fertilizer emissions and emissions factors. Their results are consistent with previously noted studies of shifting spatial distribution in NH4 deposition, for example, but provide additional valuable levels of detail. My main suggestion would be to provide some quantitative assessments of uncertainty, which I think may constitute minor revisions, as they have at least qualitatively identified the key sources of uncertainty. This and a few other minor comments are included below.

Comments: 57-58: It's not clear to me what land and fertilizer use is being referred to

[Figure]

here as distinct from the studies cited in the preceding lines.

107-119: I realize this lies somewhat outside the present paper and is likely within the work of Yu 2018, but could the authors briefly comment on how such spatial resolution was known for these distributions prior to the satellite era? Here they mention how satellites were used to determine spatial reconstructions but do not comment on any other method, which presumably would be necessary for the first half of the century, nor how such different methods have been harmonized into a single consistent dataset.

173: Here emission growth just refers to fertilizer emission, right? Not emission from livestock, which is the larger component of total emissions.

Fig 1 (a): I think it would be more clear to refer to this as loss "from" N fertilizer, not loss "to" fertilizer, since the process being described here is NH3 from fertilizer to the atmosphere, correct?

Fig 2: What drives the drop in RF from the 40s to 50s in NW, NGP and SW?

197: What do the authors mean by "reportedly" here?

Fig 4: Are the stark transition at state boundaries (e.g., west Virginia in 2005, or South Dakota in many years) something that we should interpret as physically meaningful, or is this an artifact in how some of the underlying data used for these estimates is collected at the state level? If the latter, how could this information be interpolate spatially to be more useful for geophysical modeling (such as for input to an atmospheric model)?

Fig 5: to clarify, this is NH3 emission from fertilizer, not total NH3 emission. I think it is important to re-iterate this where possible, to prevent casual readers from taking these figures out of context.

320-340: This is a very valuable list and set of discussion. Still, I wish the consideration of uncertainty could be more quantitative, even if in an approximate manner. What are the contribution of each of these factors to the total uncertainty? Do the authors believe

their total estimates are accurate to within 1%? 10%? 100% or factor of two? Please elaborate.

315 - 319: Interesting, yet a bit speculative. In the NW, biomass burning may have driven some of these trends. Without more careful study, it may be prudent to remove these speculations.

Corrections:

39: Liu 2019 was not an inverse modeling study. They estimated NH3 concentrations but not emissions. The first satellite-based top-down estimate of NH3 emissions was Zhu et al. (JGR, 118, 3355-3368, 2013).

39: ))

40: containing

73: Is "spatialized" a word? Perhaps "spatially distributed"

158: I think it's worth noting that this "data" is an estimate, which will be quite uncertain at great distances between NADP sites, particularly in the western US.

170: Northwest, whereas. . .

173, 187,. . .check throughout: NH3 subscript

220: Our

227: two

317: is heavily involved in the format of PM2.5

323: application

---

## Author Comment (AC1) · 10 Aug 2020

**Response to Review 1:**

We thank the reviewer for scrutinizing our manuscript and providing insightful comments and constructive suggestions, which greatly improve the quality of the manuscript. Please see our responses to the comments as follows.

*In this study, Cao et al. derive US NH3 emissions associated with fertilizer application from 1900 to 2015. The strength of this study lies in the use of spatially-explicit time-series for cropland distribution and fertilizer application. The authors rely on a very simple emission scheme to estimate NH3 emissions. While this is acceptable considering the goal of this study, better quantification of the role of each factors and associated uncertainties for the authors' conclusions are needed before publication can be considered.*

*General comments*

*line 130 How would application of fertilizer at emergence (early spring) for winter wheat impact the authors' conclusions*

**Reply:** We thank the reviewer for raising this insightful question. In this revised manuscript, we reconstructed the historical crop phenology and improved the N fertilizer application timing for winter wheat, fall barley, and cropland pasture to make it more reliable and reflect the real human practices. We believe this improvement solves the concern. The newly added information can be found in Methods **2.2.4 Crop phenology**, **2.2.5 Nitrogen fertilizer use dataset.**

We also added further discussions that are related to the newly added methodology. The discussion can be found in Discussion **4.3 Monthly peaks of NH$_3$ emissions shifting from 1930 to 2015**.

**Line 122 to 154.**

2.2.4 Crop phenology

We derived state-level crop phenology information from the USDA-NASS weekly crop

progress report, which recorded the fractional acreage that has reached a given crop development stage (USDA-NASS, 2018). We linearly interpolated the weekly crop progress and identified the day at which crop development was 5%, 15%, 85%, and 95% complete. We extracted the planting and harvesting dates for all major crops except for cropland pasture. For winter wheat, we also obtained the date of dormancy breaking in the early spring (green-up) from 2014 to 2016. To gap-fill the planting date of a specific crop in a given state for missing years, we grouped states by latitude and adopted the distance-weighted interpolation (Eq. 3) using the mean date of the corresponding group.

$$Date_{i+k} = \frac{Mean_{i+k} \times Date_i}{Mean_i} \times \frac{k-i}{j-i} + \frac{Mean_{i+k} \times Date_j}{Mean_j} \times \frac{j-k}{j-i} \qquad (3)$$

Where *Date* refers to the date of a given crop development stage that contains missing values, *Mean* refers to the mean date of the given stage of grouped states, the year $i$ and $j$ are the beginning and ending year of the gap, respectively, and the year $i+k$ is the kth missing year.

The survey periods of crop progress provided by USDA-NASS vary across crops and states. For example, the data of durum wheat is available only in the years 2014 and 2015, while the data of barley started from 1996. The records of the other seven crops are available since the 1980s. To extend the crop-specific planting date records back to 1900, we adopted the approach used in the Environmental Policy Integrated Climate (EPIC) crop model, which considers daily heat unit accumulation (HU, Eq. 4) and heat unit index (HUI, Eq. 5) for crop phenological development estimation. It assumes that crops are ready to be planted or to break dormancy when the mean of daily maximum and minimum temperature equals to the base temperature (Tb) (i.e. when HU reaches 0), and to be harvested when the cumulative HU equals to potential heat units (PHU) (i.e. when HUI reaches 1). Based on the days at which 5%, 15%, 85%, and 95% crop development were completed between 1980-2015, we calculated the crop-specific Tb and PHU of each state with daily maximum and minimum temperature smoothed by a seven-day moving window from 1979 to 2015 for four percentages respectively. Instead of using the temperature at planting in fall as Tb, we used the temperature at green-up

in early spring as Tb for winter wheat and fall barley to obtain a more accurate estimation of harvesting dates of these two crops. The averages of Tb and PHU in the earliest five available years of each crop type in each state were applied to Eq. 4 and Eq. 5 to calculate the dates of all four developments of all stages for missing years back to 1900.

$$HU_k = \frac{Tmax_k \times Tmin_k}{2} - Tb_c, \quad HU_k > 0 \tag{4}$$

where *HU* is heat unit, *Tmax* and *Tmin* are daily maximum and minimum temperature in °C, *Tb* is the crop-specific base temperature in °C, *k* refers to the day k, *j* refers to crop type j.

$$HUI_i = \frac{\sum_{k=1}^{i} HU_k}{PHU_j} \tag{5}$$

Where *HUI* is the heat unit index, which ranges from 0 at planting for spring-planted crops and at green-up for fall-planted crops to 1 at harvesting. *PHU* is the potential heat units required for harvesting, *i* and *k* are day i and day k, *j* refers to crop type j.

**Line 177 to 180.**

For winter wheat and fall barley, we allocated the use of N fertilizer after planting to the green-up stage in the following year. While for cropland pasture, we adopted the application timing strategy from Goebes et al (2003), in which 1/30 of the total N fertilizer amount is applied in January, February, October, November, and December, 1/12 in applied in May, June, July, and August, and 1/6 is applied in March, April, and September.

**Page 12, line 364 to 365.**

Whereas farmers in the Southern Great Plains prefer to apply most of N fertilizer after planting for cotton and a considerable amount of N fertilizer at green-up for winter wheat, resulting in peaks in summer and early spring.

*line 305 relationship with wet deposition is not very compelling. As noted by the authors there are a lot of different factors that could be at play. I would suggest to focus on spring and fall months where the authors expect the fertilizer contribution to be maximum*

**Reply:** We agree with the reviewer that focusing on spring and fall would strengthen the association between fertilizer-induced $NH_3$ emission and $NH_4^+$ deposition. However, the only $NH_4^+$ deposition maps that are available from the National Atmospheric Deposition Program are at an annual basis. To make a comparable analysis, we here used yearly $NH_3$ emission estimation rather than the seasonal estimation. According to Pearson's correlation table, we highlighted the pixels with a significance level of 0.01 and 0.001 respectively to examine the relationship between $NH_3$ emission and $NH_4^+$ deposition in the past 31 years. The result shows that the pixels with a significance level of 0.001 concentrated in the Northern Great Plains, Kansas, some parts of the Northwest and Minnesota, which supports our conclusion that the increase of $NH_3$ emission from N fertilizer may contribute to the $NH_4^+$ deposition trend in these regions. As the reviewer mentioned, we also discussed the roles of other factors such as forest fire and livestock played in these regions.

*Trend attribution ——————*

*I recommend the authors better quantify the relative importance of the different factors that contribute to changes in the magnitude and seasonality of NH3 emissions. I would suggest the authors perform their analysis using a climatology for a) temperature, b) fertilizer type, c) spatial crop distribution, e) crop mix*

**Reply:** We agree with the reviewer's suggestion. We designed additional simulation experiments to examine the contributions of five major factors, including temperature, cropland distribution, crop type, fertilizer rate, and fertilizer type, to long-term $NH_3$ emission. We found that N fertilizer use increase dominated the dynamic of $NH_3$ emission across the US. While springtime warming weakly enhanced $NH_3$ emission in

most regions, it had an adverse effect in the Northern Great Plains and Northwest. Changes in cropland distribution and type played complicated roles impacting $NH_3$ emissions across regions and over time. In general, the spatial cropland area change slightly increased $NH_3$ emission in the intensively managed agricultural regions like the Midwest and the Great Plains but lowered the emissions in the Northeast and the Southwest. Whereas crop type rotation decreased $NH_3$ emission in most regions. However, it is noteworthy that the minor effects of cropland distribution and rotation are due to the N fertilizer input was kept constant at the level of 1960 and the cropland area changes represent the summation of cropland expansion and abandonment across the country. We added the revision in Method **2.3 Factorial contribution assessment**, Discussion **4.2 Spatiotemporal change in NH3 volatilization,** and Supplement **6 Factorial contribution analysis.**

**Line 196 to 208.**

2.3 Factorial contribution assessment

Environmental factors and human activities have considerable impacts on the dynamics of $NH_3$ emissions. We set up five simulation experiments to quantify the roles of five major factors including temperature, cropland distribution, cropland rotation, N fertilizer type, and N fertilizer application rate, in shaping $NH_3$ emission since the 1960s (Table 1). The first simulation experiment (S1) was designed to mirror the temperature effect by keeping all other four factors unchanged at the level of 1960. We set up the rest simulation experiments (S2-S5) by adding the annual change of cropland distribution, cropland rotation, N fertilizer use rate, and N fertilizer type successively to S1. In S2, we allowed the percentage of cropland in each grid cell to change following the prescribed input data but kept the crop type within grid cells unchanged. Whereas in S3, the cropland percentage and type changed simultaneously through years. We further added annual N fertilizer use rates into S4 with N fertilizer type ratio fixed in 1960. We treated 1960 as the baseline year and run all the simulations from 1960 to 2015. The value difference between the simulated year and 1960 in S1 was calculated

to estimate the temperature effect. We calculated the differences between S2 and S1, S3 and S2, S4 and S3, and S5 and S4 to assess the impacts of cropland distribution, cropland rotation, N fertilizer rate, and N fertilizer type, respectively.

**Line 333 to 337.**

The conclusion drawn from our factorial contribution analysis shows that changes in cropland area and rotation have a minor influence on $NH_3$ emission in the nation (Fig. 7), which is primarily because N fertilizer input was kept constant at the level of 1960. Besides, the cropland area changes represent the summation of cropland expansion and abandonment across the country, resulting in a relatively small contribution to $NH_3$ emission increases.

Supplement:

**6 Factorial contribution analysis**

We set up five simulation experiments to examine the factorial contributions of temperature, cropland distribution, cropland rotation, N fertilizer type, and N fertilizer use rate to $NH_3$ emission change nationally and regionally. We calculated the difference every year between simulation experiments to assess the contribution of each factor and then averaged the difference within a decade (Table S5). The positive value in the Table S5 indicates a positive effect on $NH_3$ emission.

Supplement Table 5. Factorial contributions to $NH_3$ emission changes (Gg N year$^{-1}$) across the contiguous U.S.

| Decade | Region | Temperature | Land use | Rotation | N fer rate | N fer type |
|---|---|---|---|---|---|---|
| 1960s | US | 0.98 | -4.21 | -5.33 | 87.35 | -16.86 |
| | NE | 0.16 | -0.49 | 0.11 | 2.50 | 0.23 |
| | MD | 0.41 | -1.33 | -0.85 | 39.55 | -15.84 |
| | NGP | -0.13 | -0.38 | -0.23 | 9.22 | -2.61 |
| | NW | -0.04 | 0.03 | -0.03 | 3.60 | 0.97 |
| | SGP | 0.17 | -0.38 | -1.14 | 19.02 | -3.79 |
| | SE | 0.32 | -1.15 | -3.04 | 10.09 | 2.78 |
| | SW | 0.07 | -0.51 | -0.14 | 3.38 | 1.39 |
| 1970S | US | 0.31 | 3.05 | -8.17 | 260.46 | -40.75 |
| | NE | 0.11 | -0.76 | 0.63 | 6.00 | 0.32 |
| | MD | 0.30 | 1.07 | -1.15 | 112.17 | -29.81 |
| | NGP | -0.09 | 0.07 | 0.33 | 30.80 | -7.89 |

|       |     |       |       |       |        |        |
|-------|-----|-------|-------|-------|--------|--------|
|       | NW  | -0.05 | 0.34  | -0.04 | 11.40  | 0.94   |
|       | SGP | -0.04 | 1.04  | -1.19 | 55.61  | -12.47 |
|       | SE  | -0.03 | 1.91  | -7.19 | 33.68  | 7.88   |
|       | SW  | 0.10  | -0.62 | 0.45  | 10.80  | 0.23   |
|       | US  | 1.57  | 1.01  | -8.51 | 354.80 | 7.67   |
|       | NE  | 0.14  | -1.02 | 0.88  | 7.37   | 0.55   |
|       | MD  | 0.76  | 1.38  | -0.55 | 153.27 | -6.31  |
| 1980s | NGP | -0.03 | 0.31  | 0.59  | 47.48  | -7.53  |
|       | NW  | -0.09 | 0.24  | -0.02 | 14.69  | 3.18   |
|       | SGP | 0.00  | 0.21  | -1.31 | 73.85  | -4.25  |
|       | SE  | 0.52  | 1.29  | -8.84 | 43.12  | 20.74  |
|       | SW  | 0.26  | -1.40 | 0.76  | 15.02  | 1.22   |
|       | US  | 2.53  | -3.08 | -6.35 | 410.22 | 20.95  |
|       | NE  | 0.23  | -1.54 | 0.68  | 8.58   | 0.86   |
|       | MD  | 1.19  | 0.73  | -0.79 | 162.61 | -17.30 |
| 1990s | NGP | -0.04 | 0.42  | 1.04  | 67.83  | -4.55  |
|       | NW  | 0.02  | -0.13 | -0.03 | 19.22  | 5.86   |
|       | SGP | -0.03 | 1.12  | -2.58 | 86.86  | 4.03   |
|       | SE  | 0.76  | -1.71 | -5.04 | 47.41  | 29.95  |
|       | SW  | 0.40  | -1.97 | 0.37  | 17.71  | 2.01   |
|       | US  | 1.96  | -5.55 | -6.20 | 405.63 | 68.46  |
|       | NE  | 0.18  | -1.87 | 0.73  | 9.02   | 1.52   |
|       | MD  | 0.61  | 0.24  | -0.30 | 161.38 | -14.10 |
| 2000s | NGP | -0.16 | 0.33  | 0.92  | 81.85  | 28.16  |
|       | NW  | -0.03 | -0.38 | 0.13  | 21.10  | 11.31  |
|       | SGP | 0.09  | 1.57  | -2.99 | 78.05  | 9.34   |
|       | SE  | 0.68  | -3.51 | -4.00 | 38.35  | 28.42  |
|       | SW  | 0.58  | -1.94 | -0.69 | 15.88  | 3.75   |
|       | US  | 3.77  | -7.29 | -5.64 | 434.21 | 94.37  |
|       | NE  | 0.21  | -2.05 | 0.58  | 6.62   | 0.94   |
|       | MD  | 1.10  | 0.11  | -0.46 | 177.10 | -9.50  |
| 2010s | NGP | -0.06 | 0.39  | 2.07  | 107.16 | 53.17  |
|       | NW  | 0.01  | -0.50 | 0.56  | 23.37  | 11.63  |
|       | SGP | 0.14  | 1.10  | -0.71 | 69.74  | 8.39   |
|       | SE  | 1.70  | -3.77 | -6.58 | 34.38  | 25.65  |
|       | SW  | 0.66  | -2.57 | -1.12 | 15.83  | 3.91   |

*There are two important factors that I would like the authors to analyze in more details*

*a) planting dates The authors rely on a climatology for planting dates. However, Kucharik (2006) showed using the USDA crop report that corn planting took place ~2*

*weeks earlier in 2005 relative to 1980. This dataset is available for other crops and it would be useful for authors to assess the impact of changing planting dates over this time period.*

*There also exists simple parameterizations to estimate planting dates based on temperature/ precipitation that I would recommend the authors consider to estimate the variability in planting dates before 1979 (e.g., Bondeau (2007))*

**Reply:** We appreciate the reviewer for raising this critical question and providing the information about the data source. Based on the reviewer's suggestion, we collected the crop-specific phenology changes in planting, green-up, and harvesting data in each state back to the 1980s from the USDA-NASS weekly crop progress report (https://www.nass.usda.gov/Quick_Stats/Lite/index.php). Then we used the crop model EPIC to estimate the crop-specific phenology in each state from 1900 to 2015. Then we used this dynamic phenology data to replace our original static phenology data. This data improvement has substantially improved our estimates of $NH_3$ emission and led to inter-annual variations of monthly $NH_3$ emission due to the dynamic crop phenology introduced. We added the improvement in Method **2.2.4 Crop phenology**, **2.2.5 Nitrogen fertilizer use dataset**, and Discussion **4.3 Monthly peaks of $NH_3$ emissions shifting from 1930 to 2015**. Please refer to our replies to the first comment raised above.

*b) could the authors comment on the impact of long-term acidification that has been reported in several studies*

*Veenstra, J.J. and Lee Burras, C. (2015), Soil Profile Transformation after 50 Years of Agricultural Land Use. Soil Science Society of America Journal, 79: 1154-1162. doi:10.2136/sssaj2015.01.0027*

*Fuqiang Dai, Zhiqiang Lv, Gangcai Liu. (2018) Assessing Soil Quality for Sustainable Cropland Management Based on Factor Analysis and Fuzzy Sets: A Case Study in the Lhasa River Valley, Tibetan Plateau. Sustainabil-ity 10:10, pages 3477*

**Reply:** We appreciate the reviewer's suggestion and references. We added the discussion in the section **4.2 Spatiotemporal change in the $NH_3$ emissions** to address

the impact of long-term soil acidification on NH$_3$ emission.

**Line 354 to 357**

Although soil acidification through long-term agricultural land use may offset the effects of the increasing use of urea-based fertilizer, more effective policies and agricultural management are still needed in those high NH3 loss proportion regions (Veenstra and Lee, 2015; Dai et al., 2018), which can prevent air quality deterioration and enhance crop NUE.

*Comparison with other inventories ———————————*

*the authors need to compare their inventory against other efforts to develop historical emissions from EPA, EDGAR, and CMIP6. I believe that only gridded NH3 emissions from agriculture may be readily available from EPA and CMIP6 but I encourage the authors to contact the inventories' developers to obtain their estimates for historical US fertilizer emissions.*

*http://www.globalchange.umd.edu/ceds/     ->     code     is     freely     available https://edgar.jrc.ec.europa.eu/*

**Reply:** We appreciate the suggestion to show more comparisons with other NH$_3$ emission inventories and the inventory sources provided. Since our study focuses specifically on the NH$_3$ emission from synthetic nitrogen fertilizer, we cautiously chose the inventories which are comparable to valid the spatiotemporal and monthly pattern of NH$_3$ emission in our results. The CMIP6 GCM provided estimates of NH$_3$ emission from the agricultural sector in the US based on the emission factor calculated by EDGAR (Hoesly et al., 2018). Both CMIP6 and EDGAR have a solid methodology and database in estimating NH$_3$ emission globally and regionally. However, their estimates of NH$_3$ emissions from agricultural soil contains NH$_3$ emitted from nitrogen fertilizer, rice cultivation, nitrogen-fixing crops, crop residues, and so on, which includes broader emission sources than our work. As a result, CMIP6 and EDGAR reported 1431 Gg N year$^{-1}$ and 1750 Gg N year$^{-1}$ NH$_3$ emission from agricultural soil in 2014, whereas our

study estimated 630 Gg N year$^{-1}$ from N fertilizer use in the same year. EPA-NEI started the NH$_3$ inventory from 1990 and published the data discontinuously. In the inventory, other nitrogen inputs like nitrogen deposition were incorporated. Meanwhile, NH$_3$ absorbed and released by the canopy is also considered in their estimation. With input data and methodology evolving, monthly NH$_3$ emissions from "Fertilizer" were available since 2008. We selected the inventory of the year 2011 and 2014 (Version 2) to compare with our estimates in Fig. 8 for annual emission, and in Fig. 9 for monthly emission.

[Figure]

**Figure 8. Comparison of annual NH$_3$ emissions. (a) Paired comparison between our result and individual research, (b) Boxes include 25-75% of NH$_3$ emission of all chosen years estimated by our study and other studies respectively, white lines are mean values, and whiskers comprise the whole range of data. NH$_3$ emission estimated by Paulot et al. (2014) represents the average of 2005-2008, we compared their estimate against our result of 2006.**

[Figure]

**Figure 9. Comparison of monthly NH₃ emission patterns between our estimate and other studies. Two typical monthly patters of NH₃ emission in this study were used. The estimate of 2004 represents the pattern when planting date is early, whereas the simulation of 2011 stands for the pattern when planting date is delayed. Two simulations using different approaches by EPA-NEI were chosen in the comparison. Grey boxes include 25-75% of monthly NH₃ emissions during 2005-2015, black lines are mean values, and whiskers comprise the whole range of data.**

We reached out to the EPA-NEI to request spatial maps of NH₃ emission. We were provided a gridded map of NH₃ emission in 2014. By comparison, we chose the image of the spatial pattern of NH₃ emission in 2011 from NEI FTP site (ftp://newftp.epa.gov/air/nei/2014/doc/2014v2_supportingdata/nonpoint/) instead of the gridded map in 2014 because the N fertilizer input used in 2011 is more comparable to our results. However, because the 2011 map is in a low resolution and hard to re-use, we listed the side-by-side comparison as Fig. S3 in the supplementary.

[Figure]

[Figure]

Supplement Figure 3. Comparison of spatial pattern of NH$_3$ emissions between our study (a) and EPA-National Emissions Inventory (b) in 2011.

*Technical comments:*

*line 30: please rephrase to more clearly separate the impacts associated with N deposition and with PM2.5*

**Reply:** We rephrased the description in section **4.4 Effects of increasing NH₃**

**emissions on wet NH$_4^+$ deposition**

**Line 374 to 390.**

4.4 Effects of increasing NH$_3$ emissions on wet NH$_4^+$ deposition

Although the intensive NH$_4^+$ in wet deposition concentrated in the central U.S., the largest increase in wet NH$_4^+$ deposition was found in the northern Great Plains and Minnesota from 1985 to 2015 (Du et al., 2014; Li et al., 2016). Our result shows that the increase of NH$_3$ emissions from synthetic N fertilizer in the Northern Great Plains, the Northwest, and Kansas was significantly correlated to the increase of NH$_4^+$ wet deposition during 1985-2015 (Fig. 9). NH$_4^+$ deposition is highly affected by local NH$_3$ emissions because NH$_3$ volatilized into the atmosphere has a very short lifetime and deposits close to the source quickly. Therefore, In addition to growing forest fire and livestock numbers (Abatzoglou and Williams, 2016), our study reveals that NH$_3$ emissions from increasing N fertilizer use played an important role influencing the inter-annual variability of wet NH$_4^+$ deposition in the northwestern U.S. over recent decades. . Whereas with decreasing NH$_3$ emissions from N fertilizer in parts of Washington, Wisconsin, Florida, the Southeast and the Northeast since 1980 (Fig. 2), the wet NH$_4^+$ deposition promoted by an increasing forest fire, rapid urbanization, and growing livestock population (Fenn et al., 2018) showed strong negative relations with NH$_3$ emissions from synthetic N fertilizer in these regions. In addition to wet NH$_4^+$ deposition, the PM2.5 also showed an increasing trend in Minnesota, the Northern Great Plains, and the Northwest during 2002 and 2013 (U.S. EPA, 2019). Since NH$_3$ in the atmosphere heavily involves in formatting PM$_{2.5}$, the increase of NH$_3$ emissions may contribute to the PM$_{2.5}$ increase in these regions. Therefore, the increase of NH$_3$ emissions induced by northwestward corn and spring wheat expansion and consequent urea-based fertilizer use might largely enhance the environmental stress in these regions.

*line 70 I would recommend discussing alternative (more recent) approaches used to derive NH3 emissions not only in the US but also in China and Europe. There have been a lot of progress in NH3 inventories since the work of Bouwman and the authors need to better explain why this approach was selected.*

**Reply:** We agree with the reviewer's suggestion for including discussions in the model selection. Our study focus specifically on $NH_3$ emission from the single source: synthetic N fertilizer. Compared to inversed model approaches and process-based models, which mix other sources of $NH_3$ emission and require a deep understanding of various $NH_3$ emission drivers, empirical model-based emission factor has been proven an effective and valid tool for estimating $NH_3$ emission. Our work builds upon a newly developed N fertilizer management dataset including the crop-specific information of N fertilizer use rate, fertilizer type, application timing, and application method. Using high-spatial-resolution soil properties, daily temperature, dynamic crop distribution, and dynamic crop phenology as model drivers, the REML developed by Bouwman et al. (2002) can provide higher levels of detailed $NH_3$ emissions over space and time. We added the discussion in Discussion **4.5 Uncertainty**

**Line 407 to 416**

**4.5 Uncertainty**

Zhou et al (2015) developed a nonlinear Bayesian tree regression model as a function of N fertilizer rate to estimate NH3 emission in China and found the estimates match well with observations and satellite-based products. Thus, we may underestimate NH3 emissions under a high N fertilizer use rate. Another example is the use of nitrification and urease inhibitors. Nitrification inhibitors have been found to increase NH3 loss while urease inhibitors can limit NH3 volatilization (Lam et al., 2017). Therefore, the uncertainty of usage of nitrification and urease inhibitor is likely to misrepresent NH3

emissions. In addition, considering the bidirectional exchange process may improve the accuracy of seasonal NH3 emission estimation (Bash et al., 2013). However, our work builds upon the newly-developed N fertilizer management and crop phenology dataset that combines crop-specific N fertilizer use rate, fertilizer type, application timing, application method, and phenology for each state ranging from 1900 to 2015. The REML model we are using makes sufficiently use of these information and provides higher levels of details over space and time.

*line 42 grammar: for quantifying long-term spatially explicit of NH3 emissions*
*line 63 objects -> goals*
**Reply:** We thank the reviewer for these words correction and corrected them.

*Line 136 The authors need to clarify that this dataset represents a climatology of present-day planting dates.*
**Reply:** We reconstructed the historical crop phenology data, please find the response above.

*line 196 I am not sure what reportedly means in this context*
**Reply:** We have deleted the word.

*Additional comment: I forgot to mention this recent study that the authors also need to consider:*
*https://onlinelibrary.wiley.com/doi/pdf/10.1111/gcb.14499*
**Reply:** We have included this work.

**Line 42 to 43.**

Process-based modeling is a popular "bottom-up" approach for quantifying spatially explicit NH3 emissions over a long period (Cooter et al., 2012; Riddick et al., 2016; Xu et al., 2018).

---

## Author Comment (AC2) · 10 Aug 2020

**Response to Review 2:**

We thank the reviewer for scrutinizing our manuscript and providing insightful comments and constructive suggestions, which improve the quality of the manuscript. Please see our responses to the comments as follows.

*The manuscript by Cao et al. estimates NH3 emissions from fertilizer in the US over the past century. By tracking different types of fertilizer and crops, they identify variability in the spatial distribution of fertilizer emissions and emissions factors. Their results are consistent with previously noted studies of shifting spatial distribution in NH4 deposition, for example, but provide additional valuable levels of detail. My main suggestion would be to provide some quantitative assessments of uncertainty, which I think may constitute minor revisions, as they have at least qualitatively identified the key sources of uncertainty. This and a few other minor comments are included below.*

*Comments: 57-58: It's not clear to me what land and fertilizer use is being referred to here as distinct from the studies cited in the preceding lines.*

*107-119: I realize this lies somewhat outside the present paper and is likely within the work of Yu 2018, but could the authors briefly comment on how such spatial resolution was known for these distributions prior to the satellite era? Here they mention how satellites were used to determine spatial reconstructions but do not comment on any other method, which presumably would be necessary for the first half of the century, nor how such different methods have been harmonized into a single consistent dataset.*

**Reply:** To reconstruct the spatially explicit cropland distribution maps that go back to 1900, we harmonized multiple state- and national-level inventory data and remote sensing products in different periods. USDA-CDL and NLCD provide the detailed spatial distribution of crop information and are directly resampled for the reconstruction of cropland maps during the recent decade. Another satellite-based database HYDE cropland maps, which were developed by assimilating both inventory and satellite data,

was used to reconstruct the spatial maps before 2000 by depicting the potential distribution of agricultural land. Meanwhile, adjusted state- and national-level crop-specific land acreage from USDA survey data was used to limit the acreage of each crop for each state on maps. We think incorporating a detailed description of the methodology of cropland distribution map reconstruction is irrelevant to this study and therefore gave a brief summary and referred to the article that elaborates the cropland maps reconstruction process.

[Figure]

Yu, Z, Lu, C. Historical cropland expansion and abandonment in the continental U.S. during 1850 to 2016. Global Ecol Biogeogr. 2018; 27: 322–333. https://doi.org/10.1111/geb.12697

Fig 1 (a): I think it would be clearer to refer to this as loss "from" N fertilizer, not loss "to" fertilizer, since the process being described here is $NH_3$ from fertilizer to the atmosphere, correct?

**Reply:** We agree with the reviewer and we have corrected it.

[Figure]

**Figure 1. Contributions of major crop types and N fertilizer types to historical NH₃ emissions since 1900. (a) Crop specific NH₃ emissions, (b) Relative contributions of 12 major N fertilizer types to annual total NH₃ emission. Solid line in (a) refers to the NH₃ loss percentage to total N fertilizer input.**

*173: Here emission growth just refers to fertilizer emission, right? Not emission from livestock, which is the larger component of total emissions.*

**Reply:** We thank the reviewer for pointing out the vague expression here. Our study focused specifically on $NH_3$ emission from synthetic fertilizer. We added the clarification to address that $NH_3$ emission in this study refers to N fertilizer-induced $NH_3$ emission.

**Line 62 to 65.**

Based on spatially explicit time-series of cropland distribution maps and N fertilizer management database, we adopted empirical modeling of EF to calculate monthly $NH_3$ emissions from synthetic N fertilizer uses (Hereafter, $NH_3$ emission refers to the synthetic N fertilizer-induced $NH_3$ emission unless specified otherwise) in the contiguous U.S. at a resolution of 1 km × 1 km from 1900 to 2015.

*Fig 2: What drives the drop in RF from the 40s to 50s in NW, NGP and SW?*

**Reply:** We agree with the reviewer's comment. The widespread popularity of Anhydrous ammonia and Ammonium nitrate occupied the share of Urea in these regions, which largely reduced the ratio of $NH_3$ emission lost from fertilizer. This phenomenon can be found in five out of seven regions. We revised the sentence in **4.2 Spatiotemporal change in the $NH_3$ emissions** to include the region into discussion.

**Line 317 to 318.**

The "V" shape of historical national and regional $NH_3$ emission factors mainly resulted from the changing preference in using different N fertilizer types (Cao et al., 2018).

*197: What do the authors mean by "reportedly" here?*

**Reply:** We have deleted the word.

---

## Author Response (AR2)

**Response to Review 1:**

We thank the reviewer for providing further helpful comments and constructive suggestions, which improve the quality of the manuscript. Please see our responses to the comments as follows.

*The authors have done a great job addressing the comments raised by both reviewers, including revising the methodology. I think the paper has greatly improved. I have a few (mostly) minor comments for the authors to consider before publications.*

*a)*

*Eq. (4)*

*I think there is a mistake. Shouldn't it be (Tmax+Tmin)/2 ?*

**Reply:** We thank the reviewer for pointing out the mistake. We have revised the equation.

*b)*

*line 145. This is an interesting approach, how do the estimated Tb/PHU compared to tabulated values (e.g., Table 1 from Bondeau (2007)).*

**Reply:** We adopted Tb and PHU to estimate the planting, green-up, and harvesting dates for unavailable years. However, the crop progress in our study differs from the crop phenological development defined in the crop model from Williams et al. (1989) and Bondeau et al. (2007). We determine the Tb and PHU based on the days at which crop development were 5%, 15%, 85%, and 95% completed within a state. Whereas the Tb and PHU in Table 1 from Bondeau et al. (2007) are estimated based on the sowing and maturity dates chosen by the farmer under essentially climatic constrains. Therefore, the "planting date" in our study is usually later than the actual sowing date and the

"growing season" is shorter than the actual phenological cycle. As a result, the Tb estimated in our study is higher than that in Williams et al. (1989) and Bondeau et al. (2007), while the PHU is smaller. For example, the Tb of corn of Iowa we estimate are 13.7 $^{\circ}$C, 14.3 $^{\circ}$C, 16.8 $^{\circ}$C, and 17.5 $^{\circ}$C at 5%, 15%, 85%, and 95% completion respectively. The PHU are 1139, 1164, 1201, and 1201 correspondingly. While the Tb and PHU of corn are 8 $^{\circ}$C and 1670 in Williams et al. (1989), and falls into 5-15 $^{\circ}$C and 1600 in Bondeau et al. (2007).

*c)*

*line 364: Is the reference for this statement Cao et al. (2018)?*

**Reply:** Yes, it is. We have added the citation for this statement.

*d)*

*This one is probably the most important.*

*Section 2.3: I am still having trouble with this. While maps may not be available for each month, observations certainly are (NADP website). I encourage the authors to look at changes in NH4+ concentration (to limit the impact of precipitation changes) instead.*

*The correlation could be done by aggregating observations within each large regions (as in Fig. 2).*

*I think you mean Fig. 10?*

**Reply:** We appreciate the reviewer's suggestion. Using $NH_4^+$ concentration significantly improves the association between $NH_3$ emission and $NH_4^+$ deposition and

thus strengthens our conclusion.

Based on monthly site monitoring $NH_4^+$ concentration data, we generate the atmospheric $NH_4^+$ concentration maps using Inverse Distance Weighting interpolation method. The relationships between $NH_3$ emission and $NH_4^+$ concentration at each pixel over the past 31 years are examined seasonally and yearly. The correlation coefficient maps from these two time scales show small difference and we put the map using annual data in the discussion section and include the map using spring data as a supplementary figure. We revised the method in section **2.3 Correlation between $NH_3$ emission and deposited $NH_4^+$ concentration** and the discussion in section **4.4 Effects of increasing $NH_3$ emissions on $NH_4^+$ deposition.**

The figure number has been revised to 10 at **line 382**.

**Line 209 to 216**

**2.3 Correlation between NH3 emission and deposited $NH_4^+$ concentration**

We obtained monthly site monitoring data of precipitation $NH_4^+$ concentration for the period 1985-2015 in the North America from the National Atmospheric Deposition Program (http://nadp.slh.wisc.edu/data/NTN/ntnAllsites.aspx). After aggregating the monthly data to spring (March-June) and full year at each site respectively, we generated the atmospheric $NH_4^+$ concentration maps using the Inverse Distance Weighting interpolation method and resampled the maps to 1 km resolution to make it comparable to our estimated $NH_3$ emission maps. The associations between fertilizer-induced $NH_3$ emission and $NH_4^+$ concentration in precipitation during 1985-2015 at each grid cell were examined using Pearson correlation coefficients with statistical significance at $p < 0.01$ and $p < 0.001$.

[Figure]

Figure 10. Correlation coefficient between annual $NH_3$ emission from N fertilizer uses and annual $NH_4^+$ concentration in precipitation between 1985 and 2015. The correlation coefficient was calculated between the two time series at each $1km \times 1km$ grid cell. ** refers to P-value < 0.01, *** refers to P-value < 0.001.

**Appendix**

**10 Correlation coefficient between spring $NH_3$ emission and deposited $NH_4^+$ concentration**

The majority of N fertilizer is applied around planting date in spring, resulting in a peak of $NH_3$ emission from March to June in the contiguous U.S. Therefore, we examined the relationship between fertilizer-induced $NH_3$ emission and deposited $NH_4^+$ concentration during this period using Pearson correlation coefficient at each 1 km pixel over 1985-2015 (Fig. S6). The spatial pattern is very similar to the correlation coefficient between the fertilizer-induced $NH_3$ emission and deposited $NH_4^+$ concentration at an annual time scale. It implies the annual-scale relationship is mainly driven by the spring season.

[Figure]

Supplement Figure 6. Correlation coefficient between spring (March-June) $NH_3$ emission from N fertilizer uses and $NH_4^+$ concentration in precipitation between 1985 and 2015. The correlation coefficient was calculated between the two time series at each $1km \times 1km$ grid cell. ** refers to P-value < 0.01, *** refers to P-value < 0.001.

*e)*

*line 375: "Although the intensive NH4+ in wet deposition concentrated in the central U.S." I am not sure what this means.*

**Reply:** We agree with the reviewer that the expression here is not clear. We have rephrased the sentence as follow. **Although the central U.S. is the hotspot of $NH_4^+$ deposition, the largest increase in wet $NH_4^+$ deposition was found in the northern Great Plains and Minnesota from 1985 to 2015 (Du et al., 2014; Li et al., 2016)**